# Massively parallel interrogation of protein fragment secretability using SECRiFY reveals features influencing secretory system transit

Morgane Boone [1,2,9✉], Pathmanaban Ramasamy [1,3,4,5,6], Jasper Zuallaert[1,2,7,8], Robbin Bouwmeester [1,3], Berre Van Moer [1,2], Davy Maddelein[1,3], Demet Turan[1,3], Niels Hulstaert[1,3], Hannah Eeckhaut [1,2], Elien Vandermarliere[1,3], Lennart Martens [1,3], Sven Degroeve[1,3], Wesley De Neve [7,8], Wim Vranken[4,5,6] & Nico Callewaert [1,2✉]

While transcriptome- and proteome-wide technologies to assess processes in protein bio-genesis are now widely available, we still lack global approaches to assay post-ribosomal biogenesis events, in particular those occurring in the eukaryotic secretory system. We here develop a method, SECRiFY, to simultaneously assess the secretability of >$10^5$ protein fragments by two yeast species, *S. cerevisiae* and *P. pastoris*, using custom fragment libraries, surface display and a sequencing-based readout. Screening human proteome fragments with a median size of 50–100 amino acids, we generate datasets that enable datamining into protein features underlying secretability, revealing a striking role for intrinsic disorder and chain flexibility. The SECRiFY methodology generates sufficient amounts of annotated data for advanced machine learning methods to deduce secretability patterns. The finding that secretability is indeed a learnable feature of protein sequences provides a solid base for application-focused studies.

[1] Center for Medical Biotechnology, VIB, Zwijnaarde, Belgium. [2] Department of Biochemistry and Microbiology, Faculty of Sciences, Ghent University, Ghent, Belgium. [3] Department of Biomolecular Medicine, Faculty of Medicine and Health Sciences, Ghent University, Ghent, Belgium. [4] Structural Biology Brussels, VUB, Brussels, Belgium. [5] Structural Biology Research Center, VIB, Brussels, Belgium. [6] Interuniversity Institute of Bioinformatics in Brussels (IB)2, ULB-VUB, Brussels, Belgium. [7] Center for Biotech Data Science, Ghent University Global Campus, Songdo, Incheon, South Korea. [8] IDLab, ELIS, UGent, Ghent, Belgium. [9]Present address: Department of Biochemistry and Biophysics, UCSF, San Francisco, CA, USA. ✉email: morganeboone@gmail.com; nico.callewaert@vib-ugent.be

The eukaryotic secretory system processes roughly a quarter of the proteome[1–3], ensuring correct folding, assembly, and delivery of proteins to the extracellular environment, the plasma membrane, or membrane-bound organelles[4–6]. Model secretory cargos such as yeast carboxypeptidase Y (CPY), α-1 antitrypsin (AAT), transthyretin (TTR), the cystic fibrosis transmembrane conductance regulator (CFTR), and the vesicular stomatitis virus G protein (VSVG) have been instrumental in understanding the function and regulation of many ER- or Golgi-resident proteins (for instance refs. [7–11]); yet, the precise features that enable or prevent secretory system transit of the thousands of other secretory proteins remain obscure. For example, it is generally unknown which chaperones are critical for assisting the folding of the different types of secretory protein domains, what sequence or structural features control ER export kinetics, or what determines glycan modification by Golgi glycosyltransferases. Studies that examine a broad range of proteins passing through the secretory system are integral to understanding how multiple processes integrate to produce the full set of secretory proteins. Unfortunately, most current approaches are unsuited to study a comprehensive range of protein folds after entry in the ER. Mass spectrometry (MS)-based proteomics is the predominant approach for the interrogation of post-translational events, but despite many technological advances, its breadth and depth is limited and decreases steeply with sample complexity; in routine MS setups, generally, less than 70% of all transcribed mammalian protein-coding genes are detected[12,13], and full protein coverage is rarely achieved.

The secretion of recombinant proteins by heterologous hosts has long been a popular alternative to cytoplasmic expression because of the more straightforward purification, even for proteins that are not naturally secreted or located in a membrane. However, obtaining detectable levels of functional recombinant protein secreted by a given heterologous host is still too often a process of trial and error. Predicting the compatibility between recombinant protein and secretory host, and the engineering of protein or host toward increased compatibility, require models of the relationship between the amino acid sequence or structural determinants and successful protein secretion. Arguably, the availability of large-scale protein secretion data will help to demystify which and why proteins fail to pass the secretory pathway. The screening of parallel constructs or variant libraries of a protein of interest to increase recombinant protein expression success rates has gained momentum[14–17], but it is often focused on intracellular expression and more importantly, concentrates on just a single target. More comprehensive strategies to assess heterologous expression across entire proteomes do exist, but have generally also been limited to intracellular expression in *E. coli*, small proteomes, and cumbersome clone-by-clone strategies[18–21]. Thus, new methods for measuring secretion in high throughput are needed.

We here develop an approach to evaluate the secretory potential ("secretability") of proteins on a proteome-wide scale. SECRiFY (secretability screening of recombinant fragments in yeast) combines yeast surface display screening of protein libraries and a deep sequencing readout, enabling the systematic identification of heterologous polypeptides that can pass (or evade) the secretory quality control checkpoints of the yeast ER, Golgi, secretory vesicles and plasma membrane, and be secreted. As a first fundamental question to be addressed using this methodology, we ask whether, given a particular sequence of a protein fragment, we could (1) learn what features contribute to its secretability and (2) generate machine-learned secretability classifiers. Hence, we fragment the human proteome and screen these fragments for secretability in two different yeast species, *Saccharomyces cerevisiae* and *Pichia pastoris* (*Komagataella*

*phaffii*), generating a large, freely accessible repository of more than 20,000 experimentally determined yeast-producible human protein fragments. We use them to train (deep) machine learning models for secretability prediction, which unveil sequence and structural determinants of productive secretory system transit, highlighting the utility of SECRiFY to provide further insight into the basic mechanisms of secretory processing. More application-focused implementation of SECRiFY (focusing on fixed-boundary protein domains or multi-domain fragments) should enable generating databases of experimentally validated secretable native protein domain fragments, potentially advancing our understanding of their specific secretory processing mechanisms. Ultimately, this could substantially speed up experimental protein expression in many fields of study.

## Results

**Normalized fragment libraries for screening at domain-level resolution**. Multi-domain proteins often fail to express or secrete in their entirety due to local issues with misfolding of particular protein areas, translation inhibitory sequences, protease susceptibility, the absence of stabilizing interaction partners or modifications, or toxicity. The structural, functional and evolutionary modularity of proteins in domains, however, implies that individual expression of certain protein parts, especially domains, can often nonetheless be achieved. Chopping up difficult proteins into experimentally tractable fragments has been exploited by structural biologists for years, both in rational target design as well as in random library screens for soluble expression[22–26]. Moreover, screening of protein domains or fragments can provide valuable information that is not immediately attainable or obvious from screens with full-length proteins[27]. Some domain-focused interactome studies, for example, have allowed immediate delineation of the minimal interacting regions and the detection of more interactors without increasing the number of false positives[28]. We thus rationalized that screening libraries of domains or domain-sized polypeptides, rather than full-length proteins, would allow for a higher resolution measurement of secretability across proteomes, and facilitate the identification of sequence or structural features contributing to secretion.

Domain boundary prediction, however, is notoriously inaccurate, and even with a reliable estimate, small variations in the exact N- and C-terminus of the fragment can lead to dramatic differences in expressability[29]. Random approaches, on the other hand, can generate libraries of fragments that encompass most domains of a proteome by sheer oversampling. We therefore designed and built directional, randomly fragmented cDNA libraries covering the human transcriptome with fragments coding for approx. 50–100 amino acids, which is the median domain size of human proteins (Fig. 1a, c). Due to the large dynamic range in abundance of mRNA transcripts in human cells (differing over 4 orders of magnitude), however, capturing the full diversity of fragments would require unfeasibly large libraries, even at 100 bp resolution (Fig. 1b). We, therefore, reduced fragment abundance differences by relying on the second-order kinetics of nucleic acid rehybridization after denaturation, and subsequent digestion with the Kamchatka crab duplex-specific nuclease (DSN)[30,31] (Fig. 1c, d). More abundant DNA species rehybridize faster and are therefore digested first; as such, even a single round of normalization substantially reduces abundance differences between DNA fragments (Fig. 1g). Crucially, this allowed us to downsize the libraries to a scale that is feasibly compatible with downstream cDNA library cloning and yeast transformation efficiencies ($+/- 5 \times 10^6$–$5 \times 10^7$).

To ensure directionality, random primers were tagged with a rare-cutter restriction site (PacI), which is distinct from the SfiI

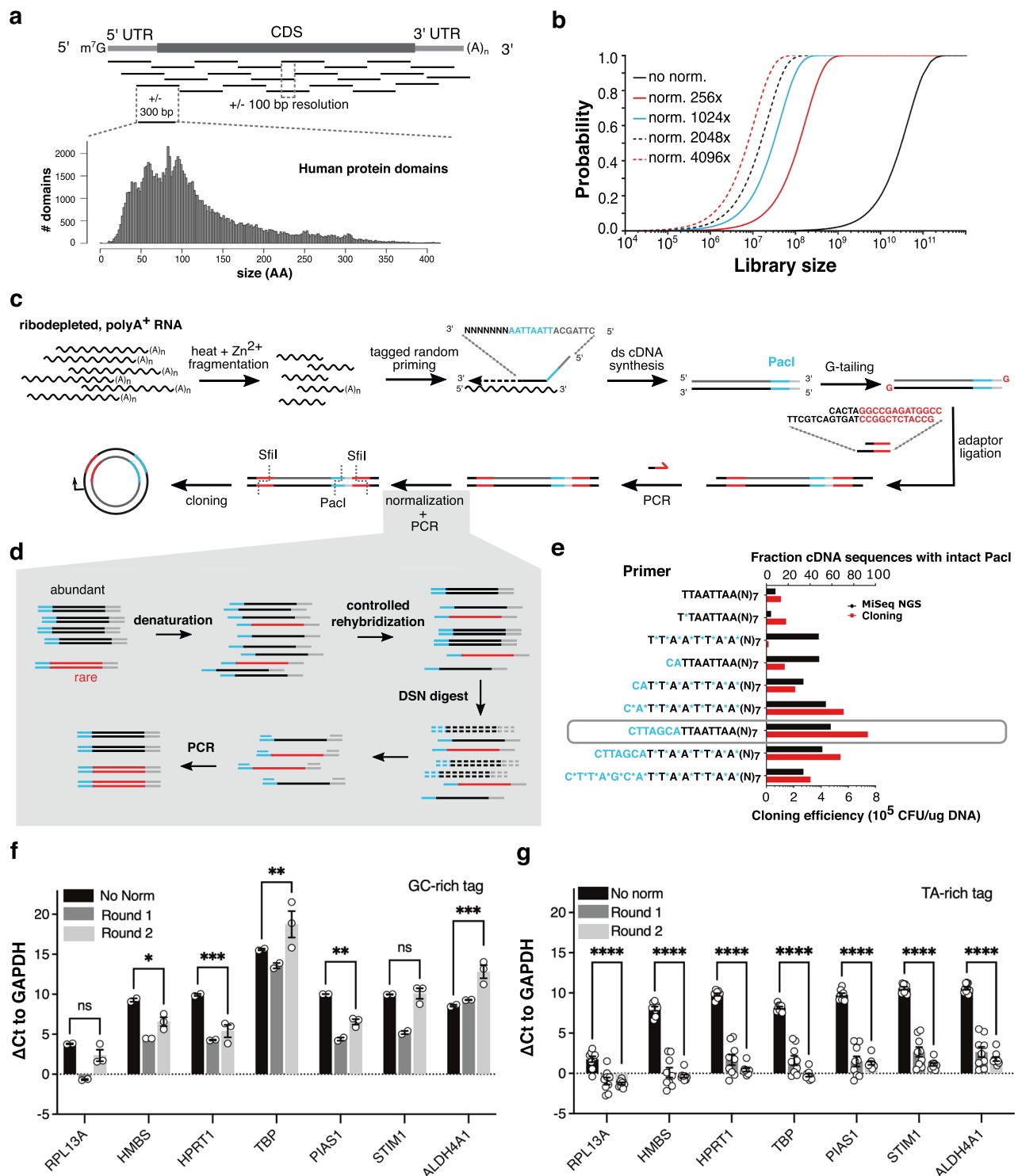

site incorporated in the library adapters (Fig. 1c). We initially observed that the random primer tag sequence is susceptible to degradation due to endo- and exonuclease activity of the *E. coli* DNA polymerase I during second strand synthesis[32–35]. As a result, less than 20% of fragment sequences contained a full-length PacI site, negatively affecting ligation into the surface display vector (Fig. 1e). Both nuclease-resistant phosphorothioate bonds and buffer sequences could partially protect the tag from degradation, and for the final library design, we settled on the primer where protection efficiency was maximal (Fig. 1e, gray bar).

Tag composition also affected abundance normalization efficiencies. In an earlier design with a GC-rich tag, normalization was less effective (Fig. 1f) than the design with PacI tag (Fig. 1g), where an ~1000-fold normalization could routinely be obtained. The tag sequence is present on all sequence fragments, and most likely, when using a GC-rich tag, rehybridization kinetics (and therefore degradation) is dominated by the tag rather than by the sequence of the fragments themselves.

In all, this library construction protocol allows for efficient capture of protein-coding fragments tiled along eukaryotic transcriptomes. It is an effective method for normalization of

**Fig. 1 Capturing protein domains from transcriptomes with directional, normalized fragment libraries. a** Most protein domains are between 50–150 amino acids (AA) long (lower left, Gene3D (v14.0.1) human protein domains, $n = 104,734$). Fragmentation of mRNA transcripts to 300 bp fragments should capture a substantial part of the domainome. At a ±100 bp resolution, on average 25 fragments would be sufficient to cover a typical transcript. **b** Estimated relationship between library size, at a hypothetical 100 bp resolution, and the probability of sampling any fragment, depending on the efficiency of fragment abundance normalization. **c** Fragment libraries are constructed by tagged random priming of fragmented polyA⁺ RNA, G-tailing, semi-single stranded adapter ligation, PCR, and duplex-specific nuclease normalization before cloning into the yeast surface display vector. ds = double-stranded. **d** Abundant transcripts rehybridize faster than rare ones during kinetically controlled rehybridization after denaturation, and as such, digestion of double-stranded DNA with duplex-specific nuclease (DSN) can be used to normalize fragment abundance. **e** Effect of phosphorothioate bonds (blue stars) and buffer sequences (blue nucleotide sequence) on degradation of the PacI sequence in the tag, as measured by deep sequencing (black bars, upper axis) and restriction enzyme/ligase-based cloning into the surface display vector (red bars, bottom axis). The design with buffering bases alone (gray box) was the most effective. Primers are written from 5′ to 3′. CFU = colony-forming units, NGS = next-generation sequencing. **f**, **g** Abundance differences of various gene fragments compared to *GAPDH*, presented as mean ΔCt ± SEM. All sequence abundances are nearly equalized when using a TA-rich (**g**), instead of a GC-rich (**f**) tag in the random primer, with normalization efficiencies up to a ±1,000-fold (ΔCt of 10) for *HPRT1*. Two-way ANOVA with Tukey post-hoc, ns: non-significant, *$p < 0.05$, **$p < 0.01$, ***$p < 0.001$, ****$p < 0.0001$. For **f** $n = 2$ biological replicates for No norm and Round 1, $n = 3$ biological replicates for Round 2. For **g** $n = 9$ biological replicates in all conditions. Exact *p*-values can be found in the Source data file Table 1.

tagged random-primed cDNA fragment libraries, and it should find many applications in areas where the protein-coding potential of a cell needs to be effectively covered in expression libraries.

**SECRiFY as a platform for secretability screening in yeast**. Relying on the sophisticated quality control (QC) machinery of the eukaryotic secretory system, which ensures efficient degradation of unstable or misfolded proteins before reaching the plasma membrane, we further reasoned that surface display could be used as a proxy for productive secretion, as other studies have suggested[19,20,36]. As such, once cloned into the surface display vector and transferred to yeast, library polypeptides are directed to the secretory system by an N-terminal secretory leader sequence derived from the yeast α mating factor (MFα1 prepro), and furthermore on the yeast cell wall via C-terminal fusion to the GPI-anchoring region of the *S. cerevisiae* cell wall protein Sag1 (Fig. 2a, b). Fragments for which the fragment-Sag1 fusion successfully passes (or escapes) secretory system QC without proteolytic degradation are recognized through their N- and C-terminal epitope tags (FLAG and V5, resp.), and are segregated from the rest using iterations of high-efficiency magnetic- and fluorescence-activated cell sorting (MACS/FACS) (Fig. 2b, c). Finally, fragment identification and classification are achieved by deep sequencing of fragment amplicons from the unsorted and sorted cell populations. In short, SECRiFY assesses secretability, i.e. the potential of a polypeptide to transit through the secretory system of ER, Golgi, vesicles, and PM without degradation, in a manner that is independent of the original endogenous localization of the protein of interest. For the present study, we focus on the basic principles of secretability. While in practice any protein-coding mRNA pool is compatible with SECRiFY, considering its biomedical importance and structural complexity, we here focused on the human proteome for our screens, as encoded by the transcriptome of various human cell lines.

We first benchmarked the method by building a $1.96 \times 10^6$ clone fragment library of the HEK293T transcriptome and performed triplicate screens in *S. cerevisiae*. On average, $1.76\% \pm 0.12\%$ of library cells displayed a fragment with an intact N-terminus (FLAG-tag) and intact C-terminus (V5-tag) (Supplementary Fig. 1). Accounting for a 1/9 chance of up- and downstream in-frame cloning, this means that ~15.8% of in-frame fragments are detectably displayed and hence, potentially secretable. After a 32-fold enrichment of these double-positive cells through a single round of MACS and two subsequent rounds of FACS (Supplementary Fig. 1), both pre- and post-sort population were sequenced at a per-base average coverage of minimally 150 reads. On average, $1.12 \times 10^6$ unique fragments/

replicate were detected, covering on average $26.45\% \pm 0.86\%$ of the human canonical transcriptome with at least three reads (Supplementary Tables 1–3). To assess the secretion-predictive value of the method, we picked random clones from the sorted population of a single experiment (Supplementary Fig. 2, Supplementary Table 4) and tested the secretion of their encoded fragments when not fused to the anchor protein Sag1. The N- or C-terminal tags of 18/20 (90%) fragments could be reliably detected on western blot from the growth medium, and for 16/20 (80%) fragments, both tags were recognized (Fig. 1d, Supplementary Table 5). As such, fragments displayed by sorted cells are indeed "secretable" with a high probability. We further classified fragments into those that were enriched (also referred to as secretable) and those that were passively depleted (hence, not detected as secretable) by sorting, setting a cut-off on the enrichment factor ($E factor = log2 \frac{FPTM_{sorted}}{FPTM_{unsorted}}$) at 1 and −1, respectively, reflecting a minimal 2-fold increase and decrease in normalized sequence read counts after sorting. Of 170,226 in-frame fragments commonly detected in the three experiments, 6.83% were consistently enriched in all three replicates, and 80.21% consistently depleted (Supplementary Table 6, Supplementary Fig. 3). Thus, using this metric, these screens were reproducible with an 87.03% concordance between replicates. These final stratified groups of fragments, which were concordantly enriched or depleted, will further be referred to as secretable and depleted, respectively.

Since we only performed positive selection for secretable fragments during screening, the depleted fraction contains only passively depleted fragments, and the negative predictive value is relatively low (40–73%, Supplementary Fig. 4). In light of this, interpretation of features affecting secretion must focus on those that affect secretability, and not non-secretability. However, as there are ±15 times as many fragments (data points) in this depleted set, this relatively low negative predictive value still provides for sufficient signal to allow machine learning methods to learn (see below).

Although we initially tested our method in the model yeast *S. cerevisiae*, in practice, the methylotroph *Pichia pastoris* (*Komagataella phaffii*) is an increasingly popular choice of host for recombinant protein production. Mostly, this has been attributed to this yeast's formidable capacity for high-density growth, the secretion of relatively few endogenous secreted proteins, and the availability of very tightly repressed and extraordinarily strong inducible promoters derived from the yeast methanol metabolism genes[37,38]. Key to the adaptation of SECRiFY for use in *P. pastoris* was the development of a modified protocol for high-efficiency large-scale *P. pastoris* transformation, which resulted in an

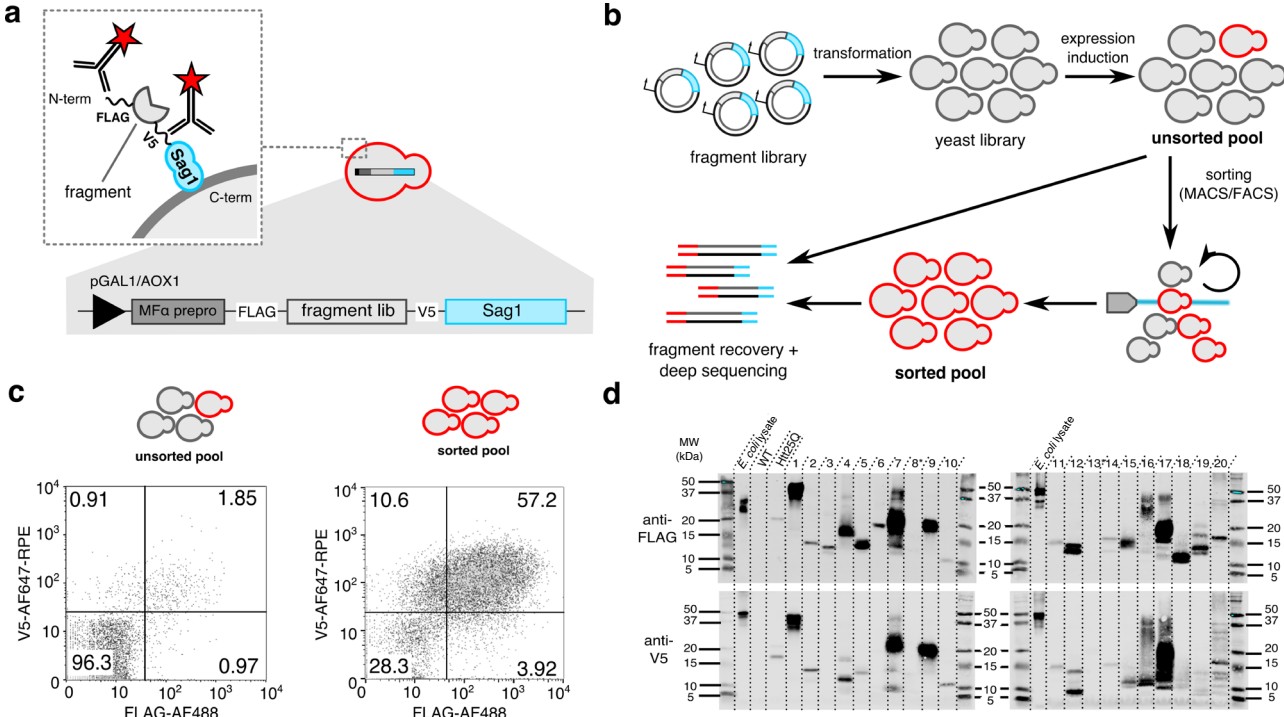

**Fig. 2 Screening for secretable protein fragments with the SECRiFY surface display platform. a** Surface display as a proxy for secretion. Libraries are cloned downstream of an inducible promoter (pGal1 for *S. cerevisiae*, and pAOX1 for *P. pastoris*), a secretory leader sequence (MFα prepro), and a FLAG tag; and upstream of a V5 tag and the Sag1 anchor. Productive passage of library polypeptide fragments through the yeast secretory system leads to incorporation into the yeast cell wall, and displaying clones are identified through antibody-based labeling of the epitope tags. **b** SECRiFY screening workflow. Fragment libraries are transformed to yeast. After fragment expression induction, displaying FLAG+V5+ clones are sorted in multiple rounds of MACS/FACS. Fragments are identified by PCR recovery and deep sequencing of both sorted and unsorted cell pools. **c** Representative flow cytometry plots for SECRiFY screening of the human proteome in *S. cerevisiae*. After 3 rounds of enrichment (MACS/FACS/FACS), the fraction of double-positive (FLAG+V5+) clones increases roughly 30-fold. **d** Western blot validation of fragment secretability after SECRiFY screening of the human proteome in *S. cerevisiae*. The majority of human protein fragments from sorted yeast cells can be expressed and secreted into the yeast medium in a Sag1-independent manner. Note that several fragments run as multiple species, likely due to heterogeneous processing and modifications such as O-glycosylation. *E. coli* lysate: antibody positive control, WT: *S. cerevisiae* R1158 medium (neg. control), Htt25Q: medium from *S. cerevisiae* secreting human Htt25Q (pos. control). Molecular weight marker units are in kDa. This experiment was performed once. Uncropped blots are provided in the Source data file.

improvement in transformation efficiency of 2–3 orders of magnitude ("Methods" and Supplementary Fig. 5). While we previously observed a slight bias toward detecting small fragments in both enriched and depleted classes in our *S. cerevisiae* pilot screen, reducing the number of PCR amplification cycles during library generation for sequencing largely eliminated this trend, although small skews occurring during both cloning and sequencing were still observed (Supplementary Fig. 6). For the *P. pastoris* screens presented here, we first generated a new fragment library with slightly larger fragment insert sizes from the pooled transcriptome of four different human cell lines (SK-N-SH_RA, GM12878, HepG2, and MCF-7) originating from diverse human tissues (brain, blood, liver, and breast), selected to maximize the number of expressed human genes based on ENCODE transcriptome data[39]. Our high-efficiency transformation to *P. pastoris* generated a library with an estimated diversity of $9.8 \times 10^6$ clones. Averaged over three replicate screens, $4.06\% \pm 0.68\%$ of cells from this library were FLAG+V5+ (Supplementary Fig. 7), which, accounting for the frequent presence of multi-copy inserts (Supplementary Fig. 8), suggests that 12.18% of in-frame fragments are displayed and hence, potentially secretable. Sequencing the fragments of unsorted cells and cells sorted after 1 round of MACS and 1 round of FACS, we detected ±1.5 million unique fragments per replicate, either in the enriched protein-displaying library, in the non-enriched starting library, or in both, covering $38.38\% \pm 2.25\%$ of the human

canonical transcriptome with at least three reads (Supplementary Tables 7–9). Of the 215,004 in-frame fragments detected in all three of the replicates, 4.84% were classified as consistently enriched, and 65.75% as consistently depleted, leading to a 71% concordance between replicates (Supplementary Table 10, Supplementary Fig. 9).

Overall, these data show that SECRiFY is a reproducible and a reliable method to estimate the secretability of protein fragments. This dataset now represents by far the largest resource on eukaryotic secretability of protein fragments.

**Secretable fragments are more flexible and disordered**. Just as cytosolic protein expression is influenced by a variety of DNA, mRNA, and protein sequence or structural features and their complex interplays[40–43], secretion of polypeptides will depend on a combination of multiple parameters, some of which are related to the unique environment and QC machinery of the ER and beyond. Even already at the simple level of general averaged parameters over our secretable vs depleted protein fragment collections, several intriguing observations emerged from our data.

We first examined whether secretable fragments differed from depleted ones in their probability to form secondary structures. To maximize the accuracy of feature prediction, fragments were filtered for size and exact match to Uniprot proteins, and condensed to an unambiguous subset of consolidated sequences

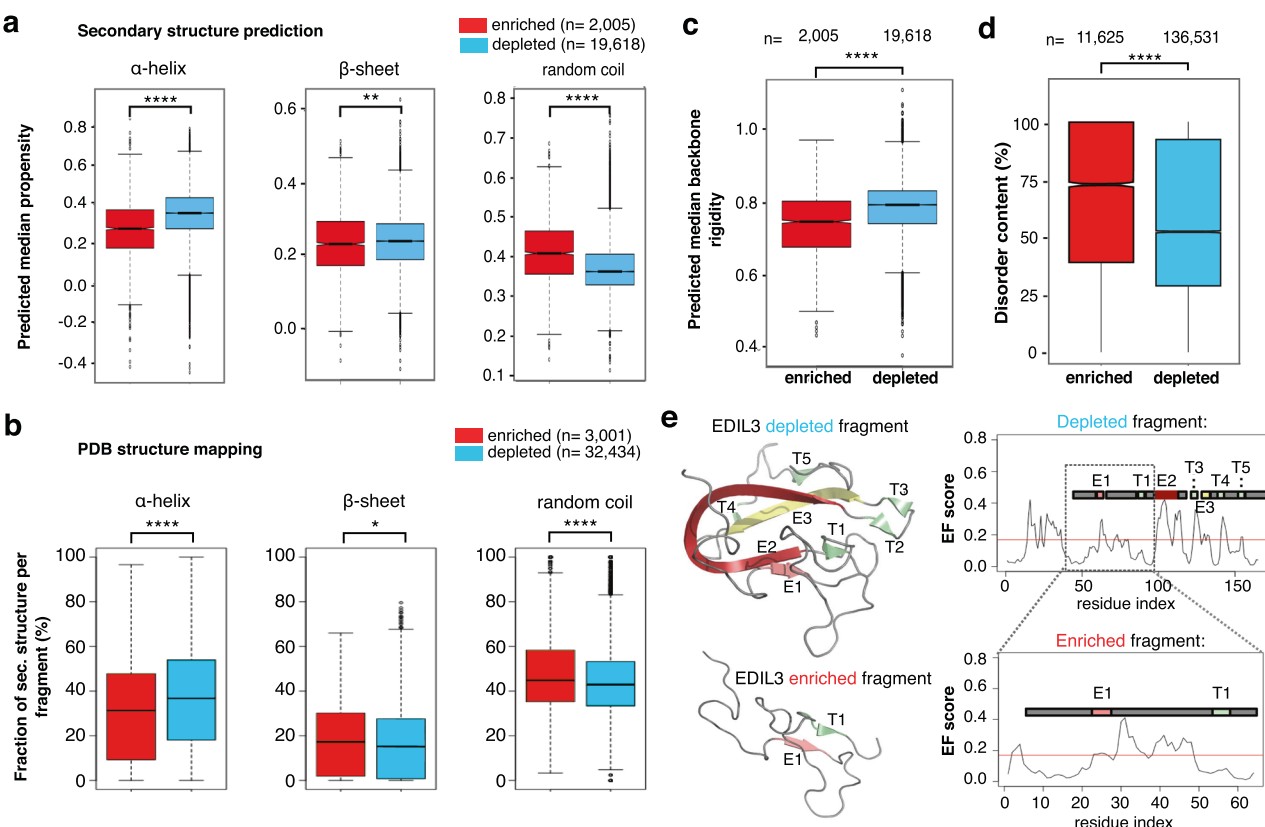

**Fig. 3 Patterns in secretable fragments. a** Dynamine predictions of secondary structure propensity in subsets of consolidated enriched ($n = 2,005$) and depleted ($n = 19,618$) fragments in *S. cerevisiae*. Enriched fragments have a lower helical content ($p = 2.95 \times 10^{-124}$) and a higher random coil ($p = 1.99 \times 10^{-127}$) propensity, which is confirmed further by mapping representative fragments to known structures in PDB (α-helix $p = 2.35 \times 10^{-8}$, random coil $p = 1.24 \times 10^{-4}$). **b** Beta sheet differences are not as pronounced ($p = 1.26 \times 10^{-3}$ for Dynamine prediction and $p = 0.02$ for PBD mapping). Enriched fragments: $n = 3,001$, depleted fragments: $n = 32,434$. **c** Enriched consolidated fragments ($n = 2,005$) are also predicted to be more dynamic than depleted ones ($n = 19,618$) ($p = 5.08 \times 10^{-118}$). **d** Similarly, the predicted disorder content in the total set of enriched fragments ($n = 11,625$) is significantly higher than in depleted fragments ($n = 136,531$) ($p < 2.2 \times 10^{-16}$). Two-sided Mann–Whitney–Wilcoxon tests in (**a–d**). *$p < 0.05$, **$p < 0.01$, ****$p < 0.0001$. **e** Two overlapping fragments of the human protein EDIL3 differ in secretability outcome. Early folding (EF) propensity predictions suggest that for the depleted fragment regions E2, T3/E3, and R4 are likely the regions driving folding of the depleted fragment, and lack of these regions in the enriched fragment result in a change in secretability. Box plots indicate the distribution of the median helix, sheet, or coil propensity of amino acid residues, summarized per (consolidated) fragment. Whiskers reflect the maximum value or the respective quartile value times 1.5 the interquartile range, whichever is less. The notch displays a confidence interval based on the median plus/minus 1.57 times the interquartile range divided by the square root of the number of points. If the notches of two boxes do not overlap, this is strong evidence that their medians differ significantly.

in order to reduce sequence redundancy (see "Methods"). Secondary structure prediction of this consolidated subset shows that secretable fragments most prominently have a lower propensity to form α-helical structures ($p = 2.95 \times 10^{-124}$, Mann–Whitney–Wilcoxon test) (Fig. 3a, Supplementary Fig. 11a). Indeed, when clustering overlapping sequences to representative fragments and mapping these to solved structures in PDB (roughly 50% of representative fragments, Supplementary Fig. 12), secretability similarly inversely correlates with α-helical content (Fig. 3b, Supplementary Fig. 13a) ($p = 2.35 \times 10^{-8}$, Mann–Whitney–Wilcoxon test). In contrast, differences in β-sheet content are only minimal (Fig. 3a, b).

Since secretable fragments also tend to more readily form random coils than depleted fragments, based on secondary structure predictions ($p = 1.99 \times 10^{-197}$, Mann–Whitney–Wilcoxon test) as well as PDB mapping ($p = 1.24 \times 10^{-4}$, Mann–Whitney–Wilcoxon test) (Fig. 3a, b), we further examined how backbone dynamics and intrinsic disorder relate to secretability. As predicted using Dynamine[44,45], secretable fragments are distinctly more flexible ($p = 5.08 \times 10^{-118}$, Mann–Whitney–Wilcoxon test) (Fig. 3c, Supplementary Fig. 14a). Disorder calculations on the full secretable vs depleted sets with RAPID[46] also confirmed a higher average intrinsic

disorder content in secretable fragments ($p < 2.2 \times 10^{-16}$, Mann–Whitney–Wilcoxon test) (Fig. 3d, Supplementary Fig. 15). In line with this, on average, fragments from both subsets appear compositionally biased. A larger fraction of secretable fragments has a higher proportion of negatively charged residues and prolines, and a tendency toward lower hydrophobicity (Supplementary Fig. 16). Possibly, this increased disorder in secretable fragments reflects how unstructured fragment sequences that lack typically exposed hydrophobic amino acids are missed by ER chaperones and can subsequently travel downstream. This is particularly striking since endogenous secretory system proteins in both human and yeast are, on average, less disordered than the whole proteome, both when considering overall disorder content ($p < 2.2 \times 10^{-16}$ and $p = 2.46 \times 10^{-5}$ resp., Fisher exact test) as well as absolute number of disordered amino acids ($p < 2.2 \times 10^{-16}$ and $p = 9.44 \times 10^{-10}$ resp., Fisher exact test) (Supplementary Tables 11–12), suggesting evolutionary counterselection.

Increasing the fidelity of the above findings, all feature enrichment observations were reproduced in the *P. pastoris* SECRiFY screens (Supplementary Tables 7–10, Supplementary Figs. 6–18). In addition, our conclusions remained unchanged when choosing alternative criteria for defining secretable vs

depleted fragment sets, illustrating the robustness of our observations.

**In silico secretability prediction with machine learning**. Our SECRiFY method generates secretability data at a scale at which training of predictive machine learning classifiers becomes feasible. To study the presence of discriminatory features in the dataset, we explored two distinct approaches: one based on feature engineering together with gradient boosted decision tree modeling[47], and a deep learning approach based on convolutional neural networks (CNNs)[48].

The gradient boosting classifier requires a fixed input size. Therefore, a series of manually engineered input features were proposed, based on physicochemical properties, sequence length, and amino acid frequencies. Ten individual classifiers were trained using different properties, and an ensemble of those was constructed using another gradient boosted classifier taking the outputs of the individual classifiers as input. The deep learning approach involved a CNN taking a one-hot encoding as input, followed by three blocks of convolutional, max pooling, and dropout layers. We explored different strategies to deal with the variable input size, as this is not supported by standard CNN architectures. A global max pooling layer yielded the best overall results. This layer is finally connected to a dense layer, followed by an output layer with a softmax.

Fragments shorter than 50 amino acids were removed from both the *S. cerevisiae* and *P. pastoris* datasets, as those are likely not long enough to properly fold, which mitigates their relevance. Using a restrictive 10-fold cross-validation scheme, where we made sure that protein fragments originating from the same gene were included in the same fold, we compared the classifiers based on the area under the receiver operating characteristic curve (AUROC). Gradient boosting achieved an AUROC of 0.781 and 0.772 on the *S. cerevisiae* and *P. pastoris* datasets, respectively, whereas the CNNs achieved AUROCs of the same magnitude, 0.779 and 0.768 (Fig. 4a). Classification results of both classifiers thus confirmed the presence of distinctive features within both secretable and depleted subsets of the data. We observed a strong correlation between the predicted values for the two approaches, with Pearson correlation coefficients of 0.810 and 0.887 on the respective datasets (Fig. 4b), which suggests that the two models learned to use similar distinctive features in the data.

Feature importance analysis using attribution methods led to compelling insights in the decisions of the CNN (Fig. 4c, d). Aggregation of individual attribution maps on the amino acid residue level indicated that the influence of individual residues on secretability is largely independent of their position in the sequence. Strikingly, there is a positive bias toward smaller residues, in line with our biophysical predictions that random coils are more readily formed in secretable fragments. Negatively charged residues also seemed to substantially contribute to secretability, confirming the pattern we picked up looking at simple averaged parameters across the full dataset. Similarly, a negative bias was observed toward all hydrophobic amino acids, affirming our earlier observations.

We further confirmed the generalizability of our prediction models by testing their performance on an independent dataset made up of fragments that were consistently enriched or depleted in solely two replicates instead of three (Supplementary Tables 15, 19). These fragments were originally excluded to maximize secretability confidence, either because they did not meet the 2-fold change threshold (set A), or because the direction of change (enriched or depleted) opposed that of the others (set B), in one of the three replicates. We now trained the gradient boosting models and CNNs on the original full dataset

(previously used for cross-validation), and evaluated on the independent data. The AUROC on set A (Sc_2consistent_1uncertain and Pp_2consistent_1uncertain) was only slightly lower than in the cross-validation, with a value of up to 0.750 (compared to 0.781 in the cross-validation) for *S. cerevisiae* and up to 0.770 (compared to 0.772 in the cross-validation) for *P. pastoris*. At the same time, predictions on set B (Sc_2consistent_1opposite and Pp_2consistent_1opposite) were shown to be less accurate, with an AUROC of up to 0.612 and 0.640, respectively. This drop in prediction accuracy is in accordance with data quality differences between both sets, as an opposite observation in the third replicate (as in set B) undermines the credibility of the label that was assigned based on the two consistent replicates. The data with a third replicate between the classification thresholds (set A) does not seem to suffer from this. In summary, these findings substantiate the value of requiring consistently enriched or depleted fragments across experiments for machine learning, and reaffirm the validity of our machine learning models.

**Secretable fragments are predicted to be enriched in certain folds and domains**. Although the results above indicate that secretable fragments are enriched for flexible or disordered chains, features that are known to affect folding in the ER conceivably could influence the secretability of those fragments that do fold. We first hypothesized that increased presence of N-glycosylation sequons or an uneven number of cysteines could favor ER retention of fragments, but we did not observe clear differences in number of Cys or N-glycosylation sequons in our datasets (Supplementary Fig. 16f, g). Furthermore, secreted and depleted fragments did not substantially differ in their predicted propensity to collapse into folded structure (EFoldMine[49] prediction, Supplementary Fig. 14b). Nonetheless, in select cases where depleted and enriched fragments overlap in sequence on the same protein, the presence or absence of regions that are most likely to fold rapidly often correlated with secretability (Fig. 3e).

Intriguingly, and despite the absence of global differences in predicted folding propensity, we did notice clear differences in predicted protein folds and domain architectures represented in both fragment groups. Of those fragments that mapped to known structures in PDB, secretable fragments are enriched in distorted sandwich and β complex folds compared to depleted fragments, suggesting that these folds are potentially more stable in the secretory environment, while the opposite is true for proteins with, for example, an α horseshoe architecture (Supplementary Table 13, Figshare links to data in "Methods"). Similarly, certain Pfam domains, such as the AAA18-domain (PF13238), are more prominent in enriched fragments than depleted fragments, while many typically cytoplasmic domains such as ribosomal protein domains or the tetratricopeptide repeat (TPR, PF13181) are found exclusively in depleted fragments (Figshare links to data in "Methods", see also Supplementary Fig. 18). This illustrates that sequence- and fold-contextual patterns of features still contain much information that is not apparent from averaged parameters.

**Protein secretability does not correlate with endogenous secretion**. Most patterns found to be enriched in secretable fragments are not recapitulated in endogenous secretory proteins. We, therefore, further evaluated whether the human proteins from which secretable fragments were derived, were enriched in secretory proteins. Since many proteins produced both secretable and depleted fragments in our screens, we considered only those proteins for which no depleted fragments were found. In both *S. cerevisiae* and *P. pastoris* screens, the proportion of secretory proteins (i.e., with a signal peptide in the endogenous setting) in

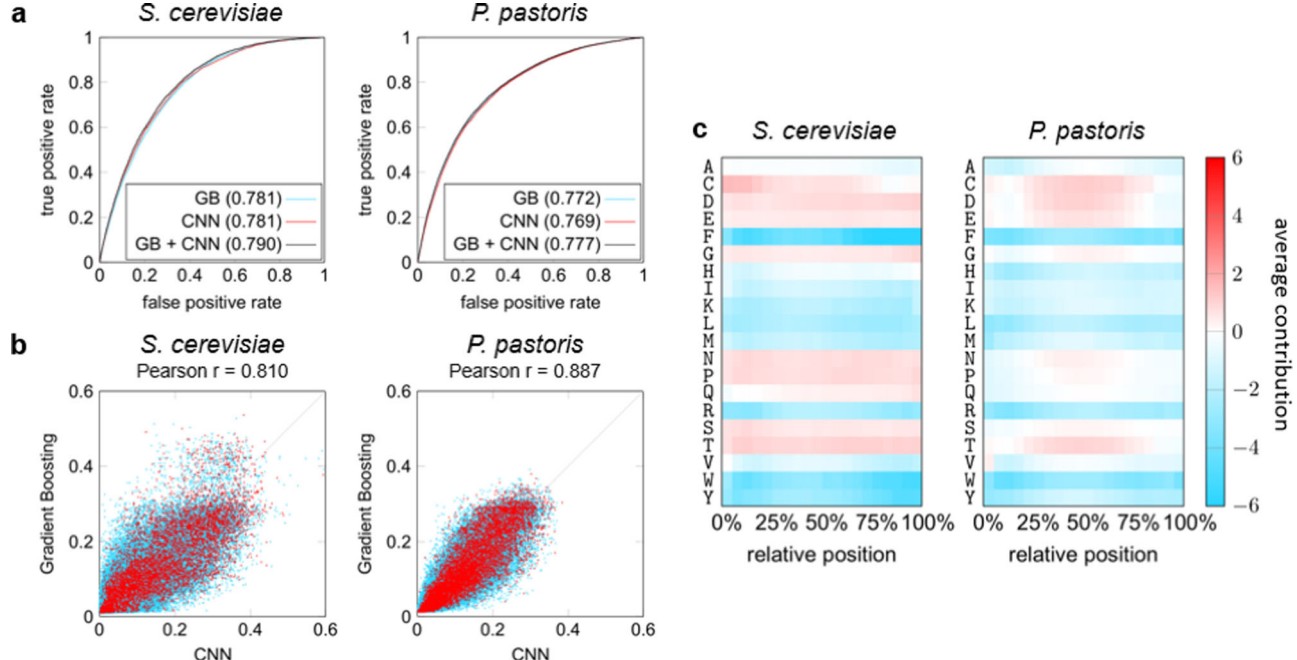

**Fig. 4 Machine learning for secretability prediction. a** Evaluation (expressed in AUROC) of the gradient boosting (GB) and convolutional neural network (CNN, one model randomly selected out of the ten trained models) approaches, as well as an ensemble taking the average predictions of both. **b** Correlation between the predicted values of the GB and CNN models. Secretable data samples are shown in red, depleted samples in blue. **c** Average contribution of individual residues when occurring in different parts of the sequence. For each sequence in a test set, the contribution toward a positive prediction (secretable) is calculated for each individual residue. Contributions are then normalized, with absolute values of all contributions in a sequence adding up to 100 on average.

**Table 1 Secretory enrichment of secretable proteins.**

| | With signal peptide | No signal peptide | p-value one-sided Fisher exact test |
|---|---|---|---|
| Human proteome | 3,574 | 16,548 | NA |
| SECRiFY secretable (S. cerevisiae) | 17 | 109 | 0.9183 |
| SECRiFY secretable (P. pastoris) | 24 | 104 | 0.421 |

this set was not significantly higher than the fraction of secretory proteins in the human proteome (Fisher's one-sided exact test, $p = 0.9183$ (*S. cerevisiae*) and $p = 0.421$ (*P. pastoris*), Table 1). This disconnect between highly secretable fragments and endogenous secretory proteins likely reflects how features that determine the highest efficiency passage through the secretory system are not the most important components of evolutionary selective pressure for secretory proteins.

## Discussion

Despite the tremendous strides made in the field of recombinant protein production, heterologous secretion remains unpredictable. A deeper understanding of the intricate ways in which different processes integrate to produce the full set of secretory system proteins, heterologous or not, is only slowly emerging. Although the study of model secretory proteins has led to substantial progress in the field, a more global approach is needed to gain a more profound and comprehensive characterization of the factors that influence secretion.

Our SECRiFY method assesses the secretability of proteins on a proteome-wide scale and at domain-sized resolution by yeast. To this end, inspired by developments in the field of massive parallel sequencing library construction and random approaches to protein engineering, we first developed a streamlined method for the construction of a directionally cloned, normalized, and random primed cDNA fragment library. This combination of features enabled us to screen the human proteome for secretability at an higher scale and depth compared to the previous studies[17,50,51]. By extension, libraries of this type could be valuable alternatives in other applications where fragment screening is beneficial, such as high throughput protein–protein interactomics.

In SECRiFY, yeast surface display screening of these libraries is combined with high-efficiency cell sorting and deep sequencing to segregate and identify protein fragments that can productively pass through the yeast secretory system. As such, we demonstrated that the secretability of protein fragments across entire proteomes can be verified experimentally in an efficient, systematic, high-throughput, and reproducible manner. Although we here used the human proteome as a testcase, our approach is generic and can be used to screen any eukaryotic, or with minor adaptations, even prokaryotic proteome. Already, the databases we have generated in this work constitute by far the largest resources of such yeast-secretable human protein segments. Remarkably, using both a gradient boosted method based on feature engineering, and an end-to-end trainable convolutional neural network approach, we achieved an AUROC of up to 0.790 for *S. cerevisiae* and 0.777 for *P. pastoris*. Practically, this means that secretable and depleted fragments have properties that allow for discrimination, even without prior knowledge of their nature. This unbiased approach confirmed our hypothesis-driven observations that biophysical features relating to secondary structure and flexibility affect secretability. Secretability is thus indeed a learnable feature of protein sequences. Ultimately, an application-focus implementation of SECRiFY will benefit recombinant protein expression and de novo protein design.

From a fundamental biology perspective, it is likely that SECRiFY will provide a means to characterize the substrate scope of secretory system processes that regulate secretory protein passage through the eukaryotic secretory system in a proteome-wide manner. This complements existing methods, such as ribosome profiling[52], which deal with protein biogenesis prior to passage through the secretory system.

Our screens, and the combined hypothesis-driven and unbiased data mining of the data, uncovered that secretable fragments tend to be less α-helical, more flexible, and more intrinsically disordered than fragments without significant display. Chaperones responsible for secretory system quality control are generally poised to recognize mostly exposed hydrophobic stretches[53–55], so conceivably, flexible yet polar and charged fragments would avoid these interactions and quickly progress toward cell wall incorporation. Limited proteolysis measurements also recently confirmed the inverse correlation of in-cell thermal stability with intrinsic disorder, α-helical secondary structure, and aspartate content[56]. Although these measurements involved proteins in their endogenous context in lysed cells, and lack of dataset overlap precluded direct comparison of secretability and thermal denaturation, it would be intriguing to investigate the relationship between secretability and stability. Even though our results show a clear correlation between disorder and secretability, for ordered proteins, quality control mechanisms in the secretory system will generally efficiently remove unfolded or unstable proteins. Indeed, recent limited proteolysis screens of known small protein domains present in PDB or Pfam suggest that structurally defined surface-displayed domains have an overall high stability score[57]. This complements our observation that secretable fragments map more frequently to certain folds or domains, and less frequently to others. Although their actual structure remains to be experimentally determined, this hints at a meaningful role of fold-contextual patterns in secretability.

Fragment detectability effects may also contribute to the observed enrichment of aspartates and glutamates. Phospho-mannose proteins of the yeast cell wall, which confer it a net negative charge, may be repelled by negatively charged displayed fragments, leading to more efficient antibody staining due to increased accessibility. Although proline-rich proteins are generally localized in the nucleus or cytoplasm[58], and are associated with ribosome slowdown[59], it is possible that the intrinsic ability of prolines to 'lock' conformations or reduce conformational freedom enhances secretory protein stability and hence, display and secretion. Polyproline or pro-rich stretches are also known motifs for binding to a wide variety of other proteins[27,60,61]; as such, the human oxidoreductase ERp57 binds calnexin and calreticulin at their pro-rich motif, and peptidyl-prolyl isomerases often act in complexes associated with multiple chaperones. Prolines are also used as gatekeeper residues against aggregation[62], and perhaps by extension, also against degradation and therefore display.

Secretable fragments are not enriched in secretory proteins. SECRiFY detects secretion at the fragment level, potentially causing some features affecting full-length protein secretion to be missed. Nonetheless, the absence of correlation also underlines that SECRiFY assesses secretability, i.e., the capacity to be secreted, rather than actual endogenous cellular localization to the secretory system. Indeed, just like most proteins are only marginally stable, endogenous secretory proteins evolved for function, not secretability.

Arguably, our method also has its limitations. The random fragmentation and size selection approach does not guarantee coverage of all possible domains, including the roughly 11% discontinuous domains found across the human proteome, but this design nevertheless allowed us to encompass a large fraction of them. More importantly, in the current SECRiFY setup, secretability was measured in the sequence context of the α mating factor prepro sequence at the N-terminus, and the Sag1 cell wall protein at the C-terminus. While results from our and other labs have indicated that for several single proteins, display efficiency correlates with relative secretion levels[63–67], it cannot be completely excluded that, at least for certain fragments, both leader sequence and the ±300 amino acid Sag1 anchor might differentially influence fragment folding, solubility, or stability. In E. coli, fusion to large proteins such as SUMO, the T. harzanium cellulose binding domain (CBD), or to maltose binding protein (MBP) is an often used strategy to promote "passenger solubilization", although again, effects vary depending on the protein[68,69]. Considering the vectorial nature of translation, a C-terminal fusion, as is the case in our setup, is nevertheless generally deemed less perturbing than an N-terminal fusion, although this is not absolute. Sag1 is also a GPI-anchored protein, affecting the entry pathway into the ER[70–72]. Similarly, the prepro leader sequence, with its multi-step processing and preference for post-translational translocation[73–75], may bias secretability of certain fragments. It remains to be determined whether similar patterns will emerge with different secretory leaders, anchors, promoters, untranslated regions, or growth conditions.

Display also imposes limitations on the dynamic range of the method, as there is an upper limit to the number of molecules that the cell wall can accommodate. Generally, this is in the range of about $10^4$ molecules per cell[76,77]. Thus, perturbations affecting secretion efficiency in these higher ranges may be missed.

In all, with SECRiFY, we here show that our fragment sequence library allowed to obtain proof-of-concept for massively parallel assessment of passage through the secretory system, providing the opportunity to learn which features influence secretability, and what rules sequences must abide by for successful transit through the yeast secretory system. We anticipate that our method and its next-generation derivatives will be of great value in both protein engineering and fundamental studies of the secretory system.

## Methods

**Plasmid construction.** All restriction digests, PCRs, plasmid preparations, and DNA purifications were performed according to the reagent/kit manufacturer's guidelines unless stated otherwise. Transformations to chemically competent E. coli MC1061 cells were done by heat shock, and cells were plated on LB agar (5 g/l bacto yeast extract, 10 g/l bacto tryptone, 10 g/l NaCl, 15 g/l agar) with the appropriate antibiotics unless noted otherwise. When working with plasmids containing the zeocin resistance cassette, E. coli TOP10 cells were used and plated on low salt LB (5 g/l bacto yeast extract, 10 g/l bacto tryptone, 5 g/l NaCl, 15 g/l agar) agar plates containing 50 μg/ml zeocin. After initial restriction digest/colony PCR/insert sequencing checkups of constructed plasmids, final plasmids were fully sequenced by the VIB Genetic Sequencing Facility using Sanger sequencing before use.

The S. cerevisiae surface display plasmid (pSSDSfiIPacI-FLAGV5-Gal1) was generated by recombination-based assembly of 3 fragments: linearized p415-Gal1-noLac as vector backbone (GAL1 promoter, CYC1 TT, CEN/ARS, LEU2 marker), a PCR product of pBluescript-ScCatch (FLAG-ministuffer-V5-Sag1), and a PCR product of the MFα1 prepro signal from pGal1-MF. PCR products were fused by overlap extension PCR, and the resulting product was recombined with linearized vector in a 30 min RT CloneEZ reaction (GenScript) and transformed to E. coli. To facilitate subsequent cloning in pSSDSfiIPacI-FLAGV5-Gal1, the small stuffer between FLAG and V5 was further replaced by a large stuffer fragment via Gibson Assembly, generating pSSDSfiIPacI-FLAGV5-Gal1-stuffer. For this, pSSDSfiIPacI-FLAGV5-Gal1 was digested with an equimolar amount of a SfiI-site containing oligo (A136) using restriction enzyme SfiI (NEB) for 50 °C. The reaction was cooled and PacI (NEB) was added, and digestion was continued at 37 °C for 1 h. Purified vector fragment was combined with the PCR amplified stuffer fragment for Gibson Assembly. Similarly, an insertless display vector, in which the Sag1 is preceded by in-frame FLAG and V5 tags, was also constructed to function as 'empty display' control for subsequent flow cytometry experiments (pSSDSfiIPacI-FLAGV5-Gal1-EV). pSSDSfiIPacI-FLAGV5-Gal1 was thus digested by BamHI/XhoI (Promega), and purified backbone was combined with amplified FLAG-V5 in a Gibson Assembly reaction. For Sag1-less secretable expression, we also constructed a vector similar to pSSD but lacking the Sag1 coding sequence. This plasmid, pSCASfiIPacI-FLAGV5-Gal1, was constructed by PCR, phosphorylation and blunt relegation. A long stuffer-containing version of this plasmid,

pSCASfiIPacI-FLAGV5-Gal1-stuffer, was constructed using the same procedure as for pSSDSfiIPacI-FLAGV5-Gal1-stuffer construction.

The *P. pastoris* surface display vector pPSDZeoSfiIPacI-FLAGV5-AOX1 was made by switching up the pPSDSfiIPacI-FLAGV5-AOX1 backbone for the pPICZ backbone through HindIII and NotI digest (both Promega), purification from gel, dephosphorylation of the pPICZ backbone, and ligation. Vector pPSDZeoSfiIPacI-FLAGV5-AOX1-stuffer was furthermore constructed by inserting part of the sequence for α-galactosidase from pPICZαGalMycHis between FLAG and V5 using SfiI/PacI restriction digest and ligation. An insertless display vector, in which the Sag1 is preceded by in-frame FLAG and V5 tags, was also constructed to function as 'empty display' control for subsequent flow cytometry experiments (pPSDZeoSfiIPacI-FLAGV5-AOX1-EV).

**Yeast strains.** *S. cerevisiae* strain R1158 (*MATa URA3::CMV-tTA his3Δ1 leu2Δ0, met15Δ0*) was obtained from Open Biosystems, frozen as slant in 15% glycerol at −80 °C, and grown on SD-Ura (0.67% yeast nitrogen base w/o amino acids, with ammonium sulfate; 2% dextrose; 0.077% CSM-Ura dropout mix; 17 g agar; pH 5.8) plates unless noted otherwise.

All *Pichia pastoris* work was performed in strain GS115 (*his4*)[78], grown in YPD media (10 g/L yeast extract, 20 g/L dextrose, 20 g/L peptone) supplemented with various concentrations of zeocin and set to various pHs as indicated, and supplemented with 17 g/L agar for plates. All plates were always freshly cast or kept in the dark at 4 °C for maximum 1 week.

**Human cell lines.** HEK293T cells were cultured at 37 °C in Dulbecco's Modified Eagle Medium (DMEM) supplemented with 10% (v/v) fetal calf serum, 2 mM L-glutamine and 110 mg/l sodium pyruvate. All cells were PPLO negative during cultivation. Cells were routinely split 1/20 with trypsin/EDTA every 3 days or when reaching max 80% confluency.

The cell lines HEK293T, HepG2, MCF7-AZ, GM12787, and SK-N-SH were obtained from the VIB IRC cell bank (HepG2, MCF7-AZ, and SK-N-SH) or from the Coriell Institute (GM12787). All cells were PPLO negative throughout cultivation and were grown without antibiotics at 37 °C in 5% CO$_2$ humidified incubators. Cells were split when reaching 70% confluency (HepG2, MCF7-AZ, SK-N-SH_RA) or when reaching max. 1 million cells/ml (GM12878). HepG2 cells were grown in Dulbecco's Modified Eagle Medium (DMEM) supplemented with 10% (v/v) fetal calf serum (FCS), 2 mM L-glutamine, 10 mM sodium pyruvate, and 100 μM non-essential amino acids. MCF7-AZ, G12878 and SK-N-SH_RA cells were grown and propagated according to the UW ENCODE Cell culture SOPs (http://genome.cse.ucsc.edu/ENCODE/protocols/cell/human/Stam_15_protocols.pdf). For MCF7-AZ, this was in Eagle's Minimal Essential Medium with 10% FCS, 2 mM L-glutamine, and 100 μM non-essential amino acids; splitting with Accutase (Thermo Fisher). For GM12878, this was in RPMI 1640 with 2 mM L-glutamine, 15% FCS. For SK-N-SH_RA, undifferentiated SK-N-SH cells were grown in RPMI 1640 with 2 mM L-glutamine, 10% FCS, and 10 mM sodium pyruvate and split with Accutase (Thermo Fisher). Prior to harvesting, cells were treated for 48 h with medium containing 6 μM *all-trans* retinoic acid for differentiation to cells with a neural phenotype.

**Human cDNA fragment library construction.** Human cell line total RNA was isolated using the Innuprep RNA MIDI Direct kit (Analytik-Jena) according to the manufacturer's instructions, additionally digesting potentially remaining genomic DNA with DNase (Turbo DNA-free kit, Ambion) for 1 h at 37 °C. RNA integrity was checked on the Agilent BioAnalyzer; all samples always had a RIN of 9 or higher. For the library screen in *P. pastoris*, samples from the HepG2, MCF7-AZ, GM12878, and SK-N-SH_RA cell lines were pooled in equal amounts. Next, poly-A$^+$ transcripts were selected with the Oligotex mRNA midi kit (Qiagen) and precipitated overnight at −20 °C in 100% RNase-free EtOH (3× initial volume) with RNase-free NaOAc pH 5.2 (0.3 M final) containing RNase-free glycogen (100 ng/μl final). Poly-A$^+$-selected RNA was recovered by pelleting for 1 h at 4 °C at 14,000 × g, washing with 70% RNase-free EtOH, and resuspended in RNase-free water (Ambion). Samples were further depleted of ribosomal RNA with the Ribo-Zero Gold (human/mouse) magnetic kit (Epicentre) following the manufacturer's instructions but using up to 7.5 μg of polyA$^+$ RNA per reaction. Ribodepleted samples were then purified with the RNeasy MinElute Cleanup kit (Qiagen). The RNA was furthermore diluted to 37.5 ng/μl in 16 μl reactions, and fragmented with 1.8 μl of Zn$^{2+}$ fragmentation buffer (100 mM ZnCl$_2$ in 100 mM Tris-HCl pH 7.0) in a PCR machine with heated lid at 70 °C for 1 min 45 s. These conditions were optimized to yield fragments with a Poisson-distributed length around 500 bp. Fragmentation was stopped with 1.8 μl 0.5 M EDTA pH 8.0, and samples were pooled and purified once more with the RNeasy MinElute Cleanup kit (Qiagen). RNA quality and size distribution was monitored at each step on a 2100 BioAnalyzer using RNA 6000 pico chips (Agilent Technologies).

In subsequent steps, contamination with environmental human genomic DNA was avoided as much as possible until after the adapter ligation step. Fragmented RNA was transcribed to double-stranded cDNA using the Maxima H minus Double-Stranded cDNA synthesis kit (Thermo Fisher Scientific) according to the manufacturer's instructions but swapping the first strand random primer for our nuclease-protected PacI-tagged random primer (primer A196, Supplementary Table 20). After RNase treatment, the cDNA was purified using RNase-free DNA cleanup beads (either AMPure XP beads (Agencourt) or CleanPCR beads (CleanNA), following manufacturer's instructions) with a 1.6:1 ratio beads:sample (v/v). The cDNA was G-tailed using Pyrophage 3137 DNA polymerase exo minus (Lucigen) in a reaction with 0.2 mM dGTP and corresponding Pyrophage polymerase buffer for 30 min at 70 °C. After DNA cleanup with beads (1.8× volume), G-tailed cDNA was ligated to the SfiI-adapter (A188_F and A188_R) in 1× Rapid Ligation buffer and 30 U/μl of T4 UltraPure DNA Ligase (Enzymatics), using 100 pmoles of adapter per 60 μl reaction, for 15 min at room temperature. Samples were purified twice in DNA cleanup beads (1.6× volume). Before normalization, samples were PCR amplified using primer A141_F (final 600 nM), which hybridizes to the adapter, and 1× KAPA HiFi HotStart mix (KAPA Biosystems) by denaturation for 3 min at 95 °C, and 20 cycles of 98 °C for 20 s, 67 °C for 15 s, 72 °C for 30 s. Samples were purified with DNA cleanup beads (1.6×) and normalized with the Kamchatka crab duplex specific nuclease (DSN) (Evrogen) as in Bogdanov et al.[31]. Briefly, per 4 μl reaction, 200 ng of cDNA is mixed with DNase-free water and 1× hybridization buffer (4× stock: 200 mM HEPES pH 7.5 with 2 M NaCl), denatured for 2 min at 98 °C, and allowed to hydridize for 5 h at 68 °C in a PCR machine with heated lid. Avoiding sample cooling, the cDNA is combined with 5 μl of pre-heated 2× DSN Master buffer (Evrogen) and equilibrated at 68 °C for 10 min, after which 0.5 μl (1 DSN unit) of DSN enzyme is added, digestion then proceeds for 25 min at 68 °C. The reaction is stopped through the addition of 10 μl of preheated 2x EDTA stop solution (Evrogen), and after a brief incubation for 5 min at 68 °C, the sample is diluted with 20 μl of DNase-free water. The single-stranded sample is then PCR amplified using 10 μl of template per 50 μl reaction with 1x KAPA HiFi HotStart mix and primer A141_F (final 600 nM) (3 min at 95 °C, and 15 cycles of 98 °C for 20 s, 67 °C for 15 s, 72 °C for 30 s). A second round of normalization is performed after sample cleanup using beads (1.6×), using the same protocol (hybridization + DSN digest + PCR + bead cleanup) but allowing hybridization for 15 h and overlaying the hybridization reaction with 10 μl of mineral oil to counter evaporation. cDNA library size distribution was monitored at each step of the procedure on a 2100 BioAnalyzer using DNA high sensitivity chips (Agilent Technologies). Normalization efficiency was controlled by qPCR comparing the levels of a set of reference genes with various expression levels (GAPDH (B002 primers), RPL13A (B005), HMBS (B003), HPRT1 (B004), TBP (B009), PIAS1 (B012), STIM1 (B013), and ALDH4A1 (B014); see primer table) in non-normalized, single-round normalized, and two-round normalized samples. All samples including controls were diluted to 5 ng/μl in DNase-free water, with final 10 μl qPCR reactions containing 2.5 ng DNA, 1× SensiFast SYBR No-ROX qPCR mix (Bioline), 300 nM forward primer and 300 nM reverse primer. Reactions were run on a LightCycler 480 (Roche) with 3 min denaturation at 95 °C, followed by 45 cycles of 95 °C for 3 s, 65 °C for 30 s (ramp rate 2.5 °C/s), and 75 °C for 1 s. Melting curves were generated to check the specificity of the reactions.

**Human cDNA library cloning and plasmid library preparation.** The cDNA fragment libraries were cloned in the *S. cerevisiae* pSSDSfiIPacI-FLAGV5-Gal1 and *P. pastoris* pPSDZeoSfiIPacI-FLAGV5-AOX1 surface display vector (for the *S. cerevisiae* and *P. pastoris* screens, resp.) using SfiI/PacI restriction digestion and ligation on a preparative scale. 200 μg of vector was first digested overnight at 50 °C with SfiI (NEB) in CutSmart buffer (NEB) and an equal molar amount of SfiI-site containing oligo (A136) according to the manufacturer's protocol, in 50 μl aliquots. After cooling to room temperature, PacI (NEB) was added and digestion was allowed to proceed for 1 h at 37 °C. The backbone band was purified from agarose gel, and dephosphorylated for 1 h at 37 °C using a thermolabile alkaline phosphatase FastAP (Thermo Scientific) that was inactivated at 75 °C for 5 min after dephosphorylation. The cDNA library was also digested sequentially with SfiI and PacI, without A136 oligo, and purified with the NucleoSpin kit (or DNA Clean and Concentrator 500 kit (ZymoResearch) for larger scale purifications) and desalted using CleanPCR beads. Digested library and dephosphorylated vector were combined in a 20:1 molar ratio for ligation with T4 DNA ligase (Thermo Scientific) using the provided T4 Ligase buffer (which was aliquoted to avoid multiple freeze-thaw cycles), aliquoted in 50 μl reactions in a PCR plate, overnight at 16 °C in a PCR machine with cooled lid. Prior to electroporation, the reactions were pooled, purified over 1.4H× CleanPCR beads, eluted in purified water (3/8ths the original ligation reaction volume), and kept on ice until electroporation.

For electroporation, freshly streaked *E. coli* MC1061 (*S. cerevisiae* screen) or TOP10 (*P. pastoris* screen) cells were grown in 5 ml of liquid LB medium (5 g/l bacto yeast extract, 10 g/l bacto tryptone, 10 g/l NaCl) at 37 °C for 1 day. The stationary culture was inoculated the following morning 1/100 in fresh LB in shake flasks of appropriate size for proper aeration, and grown while shaking at 37 °C until an OD$_{600}$ of 0.5 (about 2 h). The culture was chilled on ice for at least 30 min, pelleted for 15 min at 4000 × g at 4 °C and washed twice with ice-cold sterile water (first using 1× culture volume, then 1/2×), each time pelleting for 15 min at 4000 × g at 4 °C. A last wash was done in 1/50th of the original culture volume of ice-cold sterile 10% glycerol, to resuspend the now electrocompetent cells in ice-cold sterile 10% glycerol (600 μl per 200 ml of starting culture). Electroporation was performed in pre-chilled 96-well electroporation plates (HT-200 system from BTX), using 40 μl electrocompetent cells with 2.5 μl of purified ligation reaction per well (mix well), with the Gene Pulser electroporation system (BioRad) set at 200 Ω, a capacitance of 25 μF, a capacitance extension of 125 μF,

and a voltage of 2.5 kV. Cells were immediately transferred and pooled in SOC medium (5 g/l bacto yeast extract, 20 g/l bacto tryptone, 0.5 g/l NaCl, 2.5 mM KCl, 10 mM $MgCl_2$, 20 mM dextrose set to pH 7.0) at 1 ml SOC per reaction, and allowed to recover for 1 h at 37 °C. A serial dilution of these recovered cells was plated on agar plates with the appropriate antibiotic to assess transformation efficiency, and the rest of the culture was spread on large agar + antibiotic 24.5 cm × 24.5 cm bioassay dishes (3–4 ml per dish) using plastic sterile drigalski spatulas. After 16–24 h growth in a 37 °C incubator, all the colonies were scraped from the agar and pooled. The pellet was washed with sterile water, and weighed to assess cell number and the appropriate plasmid extraction scale, as described in the manual of the plasmid extraction kit used. The plasmid library was then extracted from the bacterial cells using one or multiple NucleoBond Xtra Midi preps (Macherey-Nagel) or QIAfilter Plasmid Giga preps (Qiagen) and eluted in Tris-HCl pH 8.5. The QIAfilter Giga preps give the overall best yield and purity. All reactions and electroporations were scaled or repeated as necessary.

Library diversity was estimated assuming equally probable variants as described in Bosley et al.[79], which states that the diversity $D = D_{max} * (1 - e^{-T/D_{max}})$ with $D_{max}$ being the maximal diversity (given an infinite number of transformants), and T the number of transformants obtained. Note that this number does not reflect the probability that a randomly picked fragment is present in the library, nor does it reflect the completeness of the library, but merely the maximal diversity possible given a particular number of transformants. In the case of our human cDNA fragment libraries, we approximate $D_{max} = 5 \times 10^7$ (assuming a normalization factor of 1024 and based on a 100 bp resolution). Note that $D_{max}$ is larger in reality as fragmentation is random. For the *S. cerevisiae* screen, we obtained an estimated $2.66 \times 10^6$ *E. coli* transformants (transformation efficiency $1.21 \times 10^5$ CFU/μg vector DNA) collected from 72 large agar dishes after 216 transformation reactions, and thus calculate a diversity of $2.59 \times 10^6$ plasmid clones. For the *P. pastoris* screen, we obtained a total of ~$1.28 \times 10^7$ *E. coli* transformants (transformation efficiency on average around $10^5$ CFU/μg vector DNA used in the ligation reaction) collected from 318 large agar dishes after 1148 transformation reactions, and thus calculate a diversity of $1.13 \times 10^7$ plasmid clones.

**S. cerevisiae library generation.** The human cDNA-surface display plasmid library was transformed to *S. cerevisiae* strain R1158 using the large-scale high-efficiency LiAc/SS carrier DNA/PEG heat shock method described in the Nature Protocols paper by Gietz and Schiestl[80] (120× scale). A small fraction of cells was serially diluted, plated and grown on SD-Leu-Ura agar plates at 28 °C for 3 days to assess transformation efficiency. The rest of the cells were immediately inoculated 1/20 in liquid SD-Leu-Ura medium (6.7% yeast nitrogen base w/o amino acids, with ammonium sulfate; 2% dextrose; 0.077% CSM-Leu-Ura dropout mix; pH 5.8) in shake flasks of the appropriate size after heat shock, and transformants were selected for 48 h at 30 °C while shaking. After selection, a small aliquot of cells was serially diluted and plated on YPD plates (10 g/l yeast extract, 20 g/l peptone, 20 g/l dextrose, 17 g/l agar) for colony PCR-based assessment of selection efficiency. The rest of the library was aliquotted and frozen at −80 °C in 15% glycerol. Transformations were scaled up or repeated as necessary.

For the library used in this screen, we obtained $3.68 \times 10^6$ yeast transformants (the transformation efficiency was $3.06 \times 10^5$ CFU/μg plasmid DNA), and with a $D_{max}$ of $2.59 \times 10^6$ (the plasmid library diversity), the estimated diversity of this yeast library is thus $1.96 \times 10^6$ clones. As is customary in the field[77,81,82], to ensure recovery of virtually all clones in downstream steps, we always worked with at least 10× as many cells as the estimated library diversity.

**P. pastoris optimized transformation procedure.** Plasmids or plasmid libraries were linearized within the AOX1 promoter with MssI (NEB, Ipswich, USA), checked for complete digestion on agarose gel and purified with CleanPCR beads (CleanNA). We modified the high-efficiency *P. pastoris* electroporation protocol as described in Wu and Letchworth[83]. Briefly, cells are grown from subcultures to an $OD_{600}$ of 1.5, pelleted at room temperature at $1500 \times g$ for 5′, and resuspended in 200 ml of sterile LiAc/DTT solution (100 mM LiAc, 10 mM DTT (from fresh 1 M stock), 600 mM sorbitol, 10 mM Tris-HCl pH 7.5) per 250 ml culture. The suspension is incubated for 30′ at 28 °C with gentle shaking (100 rpm). Pellets ($1500 \times g$ for 5′ at 4 °C) are subsequently washed 3 times with ice-cold and sterile 1 M sorbitol (37.5 ml per 250 ml starting culture), and kept on ice as much as possible. The pretreated cells are finally reconstituted in 1 M ice-cold sorbitol (1.875 ml per 250 ml starting culture) and kept on ice until electroporation. For electroporation, 80 μl of pretreated *P. pastoris* cells are mixed with 100 ng–1 μg (range tested during optimization experiments) of desalted, linearized library DNA (reconstituted in MQ) in an ice-cold 0.2 cm electroporation cuvette or electroporation 96-well plate. These mixes are electroporated at 200 Ω, a capacitance of 25 μF and capacitance extension of 125 μF, and a voltage of 1.5 kV using the Gene Pulser electroporation system (BioRad, Hercules, USA), connected to a HT-200 plate handler (BTX, Holliston, USA) for high-throughput electroporations. Immediately after electroporation, 1 ml of ice-cold YPD pH 8.0 is added and cells are transferred to appropriate flasks of tubes. The $OD_{600}$ is measured before and after a 6 h recovery with incubation at 28 °C while shaking. Cells are subsequently plated onto fresh YPD pH 8.0 agar plates containing 20 μg/ml of zeocin using glass beads to ensure uniform dispersion and grown for 3 days at 30 °C. Transformation

efficiencies are calculated based on the number of colony-forming units per μg of vector DNA, corrected with the factor of growth that occurred during recovery.

**P. pastoris library generation.** We transformed the linearized large human cDNA-surface display plasmid library to *P. pastoris* strain GS115 using the optimized library transformation procedure described above, in 184 transformations using 96-well format electroporation cuvettes (BTX) with 1 μg per transformation. A small fraction of cells was serially diluted after electroporation and recovery, and plated and grown on fresh YPD pH 8.0 agar plates containing 20 μg/ml zeocin for 2–3 days at 28 °C in order to assess transformation efficiency. The rest of the cells were inoculated 1/25 in liquid YPD pH 8.0 with 20 μg/ml zeocin, and grown at 28 °C while shaking for 2 days. In order to determine the fraction of transformed cells, a serial dilution of the selected culture was plated on non-selective YPD plates and grown for 2 days at 28 °C for colony PCR. The rest of the cells was stored at −80 °C in aliquots with 15% sterile glycerol. Corrected for the 2.74× factor growth occurring during recovery, transformation efficiency was estimated at $1.23 \times 10^5$ CFU/μg DNA, thus obtaining $2,28 \times 10^7$ transformants and an estimated maximal diversity of $9.8 \times 10^6$ clones.

As for the *S. cerevisiae* library, we always worked with at least 10× as many cells as the estimated library diversity.

**S. cerevisiae cell sorting.** For the first round of sorting, $6.89 \times 10^7$ library yeast cells were resuscitated from frozen aliquots in 10 ml of SRaf-Leu-Ura (6.7% yeast nitrogen base w/o amino acids, with ammonium sulfate; 2% raffinose; 0.077% CSM-Leu-Ura dropout mix; pH 5.8) and grown for 24 h at 28 °C while shaking. The control strain with FLAG-V5-Sag1 was inoculated from plate in 5 ml SRaf-Leu-Ura and grown under the same conditions. Expression was induced at $OD_{600}=5$ in 10 ml (library) or 5 ml (control strain) SRaf/Gal-Leu-Ura (6.7% yeast nitrogen base w/o amino acids, with ammonium sulfate; 1% raffinose; 1% ultra-pure galactose; 0.077% CSM-Leu-Ura dropout mix; pH 5.8) for 24 h, again at 28 °C while shaking. Cell pellets from two 1.5 ml aliquots of induced library culture were stored at −80 °C for plasmid extraction. The remaining cells were kept on ice or at 4 °C during the entire staining procedure. Cells were first washed 3× in ice-cold wash buffer (PBS + 1 mM EDTA, pH 7.2 + 1 Complete Inhibitor EDTA-free tablet (Roche) per 50 ml buffer, freshly made and filter sterile), each time spinning down at 4 °C for 3 min at $3000 \times g$, and stained at $OD_{600} = 4$ with mouse monoclonal anti-V5 (1/500, AbD Serotec MCA2892) and/or rabbit polyclonal anti-FLAG (1/200, Sigma-Aldrich F7425) in ice-cold staining buffer (wash buffer + 0.5 mg/ml of Bovine Serum Albumin) on a rotating wheel for 45 min at 4 °C, aliquoted in 2 ml tubes. Cell aliquots were washed 2× with 2 ml ice-cold staining buffer, and secondary staining was done with goat anti-mouse AF647-RPE (1/250, Life Technologies A20990) and/or goat anti-rabbit AF488 (1/500, Life Technologies A11008) and/or anti-mouse IgG microbeads (50 μl per ml of cells, Miltenyi Biotec 130-048-401), on a rotating wheel for 45 min at 4 °C in the dark. Cells that underwent MACS enrichment were washed 2x in MACS buffer (MACS BSA stock solution (Miltenyi Biotec) 1/20 in autoMACS rinsing solution (Miltenyi Biotec) +1 Complete Inhibitor EDTA-free tablet (Roche) per 50 ml buffer, freshly made and filter sterile). MACS enrichment was performed according to the manufacturer's protocol on a single LS column. After elution, cells were pelleted for 3 min at $3000 \times g$ at 4 °C, and recovered in 350 μl staining buffer. Cell samples that were not subjected to enrichment were washed 2× with ice-cold staining buffer. All samples were filtered over 35 μm cell strainer caps before measurement. Flow cytometry and cell sorting was performed on a MoFlo Legacy sorter (Beckman Coulter) accompanied by FlowJo v10.1 for data analysis. Fluorophores were excited at 488 nm, and fluorescence was collected through 605 short pass +530/40 band pass filters (AF488) and/or a 670/30 band pass filter (AF647-RPE). Cells were gated for a uniform SSC vs FSC single-cell population, and fluorescence quadrant gates were chosen as such that, after compensation, max. 5% of cells of unstained and single stained controls appeared above the background. We sorted out roughly 350 000 MACS-enriched FLAG+V5+ cells per screen (>10× library diversity was screened), adding 9 ml of SD-Leu-Ura + Pen/Strep (6.7% yeast nitrogen base w/o amino acids, with ammonium sulfate; 2% dextrose; 0.077% CSM-Leu-Ura dropout mix; pH 5.8 + 100 U/ml penicillin and 100 μg/ml streptomycin (Thermo Fisher Scientific)) to the collected cells for recovery. Sorted cells were then grown for 3 days at 28 °C while shaking, and frozen at −80 °C in 15% glycerol aliquots.

For the second round of sorting, round 1 sorted cells and control strains were grown, induced, stained, and sorted as in the first round but omitting MACS pre-enrichment and choosing a slightly more stringent gate to increase specificity. Cells were recovered for 4 days, part of the culture was frozen as slants at −80 °C in 15% glycerol aliquots, and part of it was frozen as pellets for plasmid DNA isolation. A dilution series of these round 2 sorted cells was plated out on SD-Leu-Ura agar plates (SD-Leu-Ura + 1.7% agar) for single clone analysis. Purity of the two-round sorted cells was verified by growing ±$2.5 \times 10^7$ cells in 20 ml SRaf-Leu-Ura + Pen/Strep (100 U/ml penicillin and 100 μg/ml streptomycin) for 48 h at 28 °C while shaking, and again inducing expression at $OD_{600} = 5$ in SRaf/Gal-Leu-Ura + Pen/Strep for 24 h at 28 °C while shaking. Cells were stained as described for the first and second sorting round, data was again collected on the MoFlo Legacy flow cytometer, and analyzed using FlowJo v10.1. The entire sorting of this yeast library was independently replicated three times on separate days.

**P. pastoris cell sorting.** For the sorting of protein fragment displaying *P. pastoris* cells, $2.2 \times 10^8$ library yeast cells were resuscitated from frozen aliquots in 100 ml of buffered complex glycerol medium (BMGY) (10 g/l bacto yeast extract, 20 g/l bacto peptone, 100 mM potassium phosohate buffer pH 6.0, 1.34% yeast nitrogen base with ammonium sulfate; $4 \times 10^{-5}$% biotin, 1% glycerol) and grown for 24 h at 28 °C while shaking. The control "empty vector (EV)" strain with FLAG-V5-Sag1 was inoculated from plate in 5 ml of BMGY and grown under the same conditions. Expression was induced at $OD_{600} = 10$ after switching the medium to buffered complex methanol medium (BMMY) (10 g/l bacto yeast extract, 20 g/l bacto peptone, 100 mM potassium phosohate buffer pH 6.0, 1.34% yeast nitrogen base with ammonium sulfate; $4 \times 10^{-5}$% biotin, 1% methanol), in 25 ml for the libraries and 5 ml for the control strain. Induction was allowed for 48 h at 28 °C while shaking, spiking in methanol to 1% every 8–12 h. At this point, a few ml of culture was subjected to genomic DNA extraction for downstream sequencing using the MasterPure Yeast DNA purification kit (Epicentre) following the manufacturer's instructions. The remaining cells were then stained, keeping samples on ice or at 4 °C during the entire procedure. Cells were first washed 3× in ice-cold wash buffer (PBS + 1 mM EDTA, pH 7.2 + 1 Complete Inhibitor EDTA-free tablet (Roche) per 50 ml buffer, freshly made and filter sterile), each time spinning down at 4 °C for 3 min at $1500 \times g$, and stained at $OD_{600} = 2$ with mouse monoclonal anti-V5 (1/500, AbD Serotec MCA2892) and/or rabbit polyclonal anti-FLAG (1/200, Sigma-Aldrich F7425) in ice-cold staining buffer (wash buffer + 0.5 mg/ml Bovine Serum Albumin) on a rotating wheel for 45 min at 4 °C. Cells were washed 2× with ice-cold staining buffer, and secondary staining was done with goat anti-mouse AF647-RPE (1/250, Life Technologies A20990) and/or goat anti-rabbit AF488 (1/500, Life Technologies A11008) and/or anti-mouse IgG MACS microbeads (50 µl per ml cells, Miltenyi Biotec 130-048-401), on a rotating wheel for 45 min at 4 °C in the dark. Cells that underwent MACS enrichment were washed 2× in MACS buffer (MACS BSA stock solution (Miltenyi Biotec) 1/20 in autoMACS rinsing solution (Miltenyi Biotec) +1 Complete Inhibitor EDTA-free tablet (Roche) per 50 ml buffer, freshly made and filter sterile). MACS enrichment was performed according to the manufacturer's protocol on two LS columns. After elution, cells were pelleted for 3 min at $1500 \times g$ at 4 °C, and recovered in 2.5 ml of staining buffer. Cell samples that were not subjected to enrichment were washed 2× with ice-cold staining buffer. All samples were filtered over 35 µm cell strainer caps before measurement. Flow cytometry and cell sorting was performed on a MoFlo Legacy sorter (Beckman Coulter) accompanied by FlowJo v10.1 for data analysis. Fluorophores were excited at 488 nm, and fluorescence was collected through 605 short pass +530/40 band pass filters (AF488) and/or a 670/30 band pass filter (AF647-RPE). Cells were gated for a uniform SSC vs FSC single-cell population, and fluorescence quadrant gates were chosen as such that, after compensation, max. 5% of cells of unstained and single stained controls appeared above the background. We sorted out approximately 5 million MACS-enriched FLAG+V5+ cells per screen (a number of events >10x library diversity was screened in total). Sorted cells were spun down at $1500 \times g$ for 5 min at 4 °C, and recovered in 20 ml of YPD pH 8.0 + Pen/Strep (100 U/ml penicillin and 100 µg/ml streptomycin (Thermo Fisher Scientific)). After 12 h, zeocin was added to 20 µg/ml. Sorted cells were grown for 36 h in total at 28 °C while shaking, and frozen at −80 °C in 15% glycerol aliquots. For genomic DNA isolation, cells were recovered in YPD pH 8.0 with Pen/Strep and zeocin, and genomic DNA was extracted using the MasterPure Yeast DNA purification kit. Library sorting was independently replicated 3 times on three different days.

**S. cerevisiae deep sequencing library preparation.** Plasmid isolation of sorted and non-sorted *S. cerevisiae* yeast libraries was performed as in Whitehead et al.[82] using the ZymoPrep Yeast Plasmid Miniprep II kit (Zymo Research). Briefly, $9–20 \times 10^7$ pelleted frozen cells were resuspended in 400 µl of Solution I with 50 U Zymolyase and incubated for 4 h at 37 °C. After a flash freeze in liquid $N_2$ and thawing at 42 °C, plasmid extraction was continued as described in the manufacturer's protocol, but eluting in 30 µl of 10 mM Tris-HCl pH 8.0. Genomic DNA was digested with 60 U of exonuclease I (NEB) and 7.5 U lambda exonuclease (NEB) in lambda exonuclease buffer (NEB) for 90 min at 30 °C, followed by inactivation for 20 min at 80 °C. Library plasmids were purified from the buffer using CleanPCR beads (2× reaction volume) (GC Biotech) and eluted in 22 µl MilliQ water. Next, the human cDNA fragments on the plasmids were recovered by PCR using two pools of "frameshifting" primers in analogy to Lundberg et al.[84], so as to equalize base distribution at the first sequenced positions in order to take maximal advantage of the sequencing chip capacity. Pools of equal molar concentration were made for A247_F*x* and for A247_R*x*. PCR reactions were set up using 20 µl purified plasmid DNA, 300 nM of each primer pool, and 1× KAPA HiFi HotStart Readymix in a final volume of 50 µl, and run for 3 min at 95 °C, followed by 25 cycles of 98 °C for 20 s, 61 °C for 15 s, 72 °C for 30 s. Samples were purified using CleanPCR beads (1.6× reaction volume) and eluted in 40 µl of 0.1× TE buffer (1 mM Tris-HCl + 0.1 mM EDTA, pH 8.0). Illumina adapter sequences and barcodes were added using the NEBNext Ultra DNA library prep kit for Illumina (NEB) largely according to the manufacturer's protocol, except that the samples were purified using two rounds of 1.6× volume CleanPCR beads after adapter ligation to remove adapter dimers, and that the final PCR was performed with custom primers (A237_F and A237_R_bcx, with bcx indicating different barcodes), desalted Ultramers from IDT) and for 25 cycles. After PCR, the 500–1200 bp

fragments were purified from 2% agarose gel using the Nucleospin gel and PCR cleanup kit (Macherey-Nagel), specifically solubilizing the agarose blocks overnight at 4 °C in NT buffer to avoid fragment denaturation and reduce GC-bias. After elution in NE buffer, samples were purified a second time using CleanPCR beads (1.6× volume) and finally eluted in 25 µl of 0.1× TE buffer in DNA LoBind tubes (Eppendorf). Reasoning that the reduced complexity of the sorted fragment pool would require less depth than that of the unsorted fragments, samples were pooled in a 2.5/1 molar ratio of unsorted/sorted libraries. Concentrations were determined using Nanodrop, Qubit, and the KAPA Library Quantification kit for LC480 on an Lightcycler 480 (Roche) according to the manufacturer's instructions. Size distributions were assessed on a 12-capillary Fragment Analyzer (Advanced Analytical) with their High Sensitivity NGS kit (DNF-474, Advanced Analytical), and the BioAnalyzer (Agilent) with the DNA High Sensitivity kit (Agilent).

**P. pastoris deep sequencing library preparation.** The cDNA fragments of sorted and unsorted *P. pastoris* library were picked up from genomic DNA by PCR (500 nM A149_F, 500 nM A149_R, 1× KAPA HiFi HotStart master mix, 70 ng genomic DNA per 20 µl reaction—95 °C for 3 min, followed by 20 cycles of 98 °C for 20 s, 61 °C for 15 s, 72 °C for 30 s before cooling). PCR fragments between 300–1000 bp in length were isolated from a 2% agarose gel using the NucleoSpin Gel and PCR cleanup kit (Macherey-Nagel) and CleanPCR beads (CleanNA), solubilizing the plugs at 4 °C to avoid denaturation of AT-rich fragments, eluting in 30 µl purified water. This pool of fagments was then further subjected to a second short PCR for the addition of frameshifting bases (500 nM A247_F primer pool, 500 nM A247_R primer pool, 1× KAPA HiFi HotStart master mix, 20 µl DNA per 50 µl reaction—95 °C for 3 min, followed by 5 cycles of 98 °C for 20 s, 61 °C for 15 s, 72 °C for 30 s before cooling) and was purified with CleanPCR beads (1.6:1 ratio beads:reaction volume) and eluted in 45 µl of purified water. Illumina sequencing library construction was done with the NEBNext Ultra DNA library prep kit (NEB) largely according to the manufacturer's protocol, except that the samples were purified using one rounds of 1.2× volume CleanPCR beads after adapter ligation to remove adapter dimers, and that the final PCR was performed with custom primers (A237_F and the barcoded A237_R_bcx, desalted Ultramers from IDT) and for 7 cycles. This number of PCR cycles was found to be optimal after a prior optimization experiment in which we followed the PCR reactions in real time in a qPCR with SYBR Green, to determine the maximal number of cycles until an amplification plateau is reached. Fragments were purified using CleanPCR beads (0.7× volume) and eluted in 25 µl of 0.1× TE buffer in DNA LoBind tubes (Eppendorf). To increase sample yields, we did an additional 4-cycle PCR with primers against the P5 and P7 sequences (500 nM A240_F, 500 nM A240_R, 1× KAPA HiFi HotStart master mix, 2.5 µl DNA per 100 µl reaction—95 °C for 3 min, followed by 4 cycles of 98 °C for 20 s, 63 °C for 15 s, 72 °C for 30 s before cooling). Fragments were again purified using CleanPCR beads (0.7× volume) and eluted in 30 µl 0.1× TE buffer in DNA LoBind tubes (Eppendorf). Samples were pooled in a 4.3/1 molar ratio of unsorted/sorted libraries. Concentrations were determined using Nanodrop, Qubit, and the KAPA Library Quantification kit for LC480 on a Lightcycler 480 (Roche) according to the manufacturer's instructions. Size distributions were assessed on a 12-capillary Fragment Analyzer (Advanced Analytical) with their High Sensitivity NGS kit (DNF-474, Advanced Analytical).

**Illumina sequencing, read processing, and sequencing data analysis.** For each screen, the pooled sample was paired-end sequenced ($2 \times 150$ bp) on an Illumina NextSeq 500 mid-throughput or high-throughput (*S. cerevisiae* or *P. pastoris* screen, resp.) chip and demultiplexed using the NextSeq System Suite 2.0.2. Raw demultiplexed Illumina sequencing data were processed using a combination of publicly available tools and custom scripts. Raw reads were first trimmed with Trim Galore! version 0.4.1 (www.bioinformatics.babraham.ac.uk/projects/trim_galore) to remove Illumina adapter sequences. Next, FLAG/V5 and frameshifting sequences were trimmed off with Cutadapt version 1.10 (ref. [85]), discarding all untrimmed pairs to only keep correctly cloned cDNA fragments. Quality control of raw and processed fastq files was performed using FastQC version 0.11.3 (www.bioinformatics.babraham.ac.uk/projects/fastqc). Processed reads were mapped to the human transcriptome of known protein-coding genes as downloaded from Ensembl's BioMart[86] using BBMap v35.40 (sourceforge.net/projects/bbmap). Count tables were built and analyzed from the properly paired mapped reads using SAMtools[87] v1.2 and v1.3, BEDtools[88] v2.24.0 and v2.25.0, EMBOSS[89] v6.6.0, R project 3.3.0 (www.R-project.org) and the R packages plyr (v1.8.6), ggplot2 (v3.3.4), alakazam (v1.1.0), stringr (v1.4.0), and UpSetR (v1.4.0)[90]. A summary of the most important scripts can be found on Figshare (figshare.com/s/5dba6b512-fa74ef68a40). Fragments were considered detected when fragment count >0 in either the unsorted sample, or the sorted sample. Enrichment factors (E factors) were calculated as $\log_2\left(\frac{FPTM_{sorted}}{FPTM_{unsorted}}\right)$, with FPTM being our custom Fragment count Per Ten Million fragments which is defined as the number of read pairs with the same start and end position per 10 million read pairs. For the concordance calculations, $FPTM_{unsorted}$ was calculated over the merged replicate unsorted samples, and from the fragments detected in all 3 replicates (sorted sample or merged unsorted), only the fragments that were in-frame with both the N-and C-term fusion parts in the surface display construct were considered (as we used random

priming, there is an expectable 1/9 chance that a cloned fragment is in the same reading frame with both the N-and C-term fusion parts).

**Flow cytometry of randomly picked sorted *S. cerevisiae* clones.** To assess the correlation between sequencing count and surface display fluorescence signal, 47 two-round sorted *S. cerevisiae* single clones and the control strain with FLAG-V5-Sag1 were inoculated in 2 ml of SRaf-Leu-Ura in deep 24-well plates and grown for 24 h at 28 °C while shaking. Cells were pelleted at 4 °C at 3000 × *g* for 3 min, supernatans was removed, cells were resuspended in 2 ml SRaf/Gal-Leu-Ura, and induced for 24 h at 28 °C. Cell staining was performed as done for the library round 2 sorts, without MACS enrichment, but working in 96-well V-bottom Nunc microwell plates (Thermo Fisher). Cells were finally diluted ¼ in staining buffer, and measured on an LSR-II HTS flow cytometer (BD). Fluorophores were excited at 488 nm, and fluorescence was collected through 550 long pass + 525/50 band pass filters (AF488), and/or 670 long pass + 685/35 band pass filters (AF647-RPE). Compensation, gating and further data analysis were done in FlowJo v10.1.

For identification, the same clones were subjected to a colony PCR targeting the human cDNA fragment that each clone encodes. For each clone, a single colony was picked from plate and resuspended in 20 µl of freshly-made 20 mM NaOH and incubated for 5 min at room temperature (RT). Lysis was stopped by adding 80 µl of sterile water, and 5 µl was used in a 25 µl PCR reaction with 0.5 U Phusion High Fidelity polymerase (NEB), 500 nM of forward primer A207_F, 500 nM of reverse primer A221_R, 1× Phusion HF buffer, and 200 µM dNTPs (Promega). PCR cycling conditions involved a 98 °C denaturation for 30 sec; 30 cycles of 98 °C for 10 s, 52 °C for 15 s, 72 °C for 45 s; finishing off with 5 min at 72 °C before cooling. PCR fragments were purified by CleanPCR beads (GC Biotech), Sanger sequenced from both ends using primers A149_F and A149_R, and obtained sequences were mapped to the human reference transcriptome sequences using BLAST and reconstructed from there. NGS fragment counts in background and sorted cell libraries were obtained by searching the count tables for fragments with the same gene symbol and amino acid sequence.

**Secretion validation via western blot.** For the validation of fragment secretability, 20 random single *S. cerevisiae* clones from the replicate 1 screen sorts were grown in 2 ml of SD-Leu-Ura, and plasmids were isolated with the ZymoPrep Yeast Plasmid Miniprep II kit (Zymo Research) according to the manufacturer's instructions. For each clone, the encoded cDNA fragments were isolated via PCR with 300 nM of A262_F and 300 nM of A262_R primers (with homologous overhangs for downstream cloning in pSCA-stuffer), in a 1× KAPA HiFi PCR reaction using 4 ng of plasmid DNA per 20 µl reaction. Samples were denatured for 3 min at 95 °C, and cycled 25× at 98 °C for 20 s, 57 °C for 15 s, 72 °C for 20 s, before cooling. Amplified DNA was extracted from gel and purified. Secretion vector pSCASfiIPacI-FLAGV5-AOX1-stuffer was digested with SfiI and PacI (both NEB), and the vector backbone was also isolated from gel and purified. Fragment and backbone were assembled using Gibson Assembly for 30 min at 50 °C, and transformed to *E. coli*. Plasmids were verified by sequencing. These 20 pSCA-fragment vectors were transformed to yeast using the LiAc/PEG method by Gietz and Schiestl[91], transformed clones were checked by colonyPCR as described above, but using primers A221_R and A221_F. For secretion induction, these single yeast clones were first grown in SRaf-Leu-Ura for 48 h at 28 °C while shaking, pelleted, and induced in SRaf/Gal-Leu-Ura for 24 h at 28 °C. Medium was collected and frozen at −20 °C until protein extraction.

Secreted proteins were pelleted from the medium by precipitation with DOC and TCA. Briefly, for each sample, 10% of the sample volume of 5 mg/ml deoxycholate (DOC) was added, the sample was incubated on ice for 10 min, 13.54 M trichloroacetic acid (TCA) was added at 10% sample volume, the sample was incubated on ice for 20 min, and the precipitate was pelleted at 4 °C in a centrifuge at max. speed for 30 min. Supernatans was removed, and the pellet was washed twice with ice-cold acetone, and once with 70% ethanol, each time pelleting the sample for 20 min at 14,000 × *g* at 4 °C. The pellets were dried at 37 °C and resuspended in 1× phosphate-buffered saline PBS. Total protein concentration was estimated with the microBCA kit (Pierce) according to the manual's instructions. For each sample, 10 µg of protein was additionally PNGase F digested (NEB) overnight, according to the manufacturer's protocol. Finally, equal amounts of protein for each sample were denatured in 1× Laemmli buffer (10% glycerol, 0.1% DTT, 63 mM Tris-HCl pH 6.8, 2% SDS, 0.0005% bromophenol blue) for 10 min at 98 °C, run on a 15% Tris-Glycine SDS-PAGE gel, and semi-dry blotted for 1 h30 on PVDF membranes at 75 mA per 45 cm² blot. Blots were blocked with 3% milk powder solution for 2 h at RT or 4 °C overnight and stained with polyclonal rabbit anti-FLAG antibody (1/2000, Sigma, F7425) + anti-rabbit IgG-Dylight800 antibody (1/15,000, Thermo Scientific, #35571), or mouse anti-V5 monoclonal antibody (1/3000, AbD Serotec, #MCA1360) + anti-mouse IgG-Dylight8000 (1/15,000, Thermo Scientific, #35521). The ladder was the BioRad Precision Plus Dual Xtra ladder. Blots were imaged with the Li-Cor Odyssey system.

**Feature enrichment analysis.** Protein and fragment structural disorder prediction was done using RAPID[46]. To assess whether secretable fragments were more likely to be derived from endogenously secreted proteins than by chance, human proteins and human secretory proteins (ie with signal peptide) were downloaded from

Uniprot (release 2018_11) and intersected with the lists of secretable and depleted fragments. Only proteins for which no depleted fragments were found were retained for analysis. For analysis of N-glycan sequons, we evaluated the presence of the sequon NXS or NXT but not NPS or NPT using custom awk code.

**Structural bioinformatics.** For the biophysical predictions, sequences were first filtered for a 100% sequence match to the UniProt protein and a length longer than 30 amino acids. Secondary structure (a-helix, b-sheet, and random coil) and early folding propensities were predicted as described for EFoldMine[92], but only retaining residues in the full protein sequence that are unambiguously 'secreted' or 'depleted' across overlapping fragments. From this, contiguous regions of secreted or depleted residues were assembled into consolidated fragments, onto which the average of all predictions for that fragment from the original fragment is condensed. Backbone dynamics of sequenced fragments were predicted using Dynamine[44,45]. Plots were generated in R (www.R-project.org) using custom Python scripts.

For PDB mapping, protein fragment sequences were first clustered into representative fragments using the CD-HIT package[92] with an identity parameter of 100%. This clusters all shorter sequences with fully overlapping longer sequences into a single longer representative fragment. The representative fragments from each dataset were used as queries to perform a blast against PDB database using standalone blast (ncbi-blast-2.6.0+). The percentage of secondary structural elements for each fragment with a PDB hit was calculated from its corresponding DSSP coordinates. Domain architectures (Pfam and Gene3D) were retrieved using InterProScan[93] (v 5.24-63.0). Frequency of a particular domain in a dataset was obtained by removing duplicate entries (if a particular domain is present more than once for a particular fragment) in the dataset.

**Statistical hypothesis testing.** Comparison of library normalization efficiencies was done using a two-way ANOVA with Tukey post-hoc test. Screen replicability was assessed through calculation of Spearman correlation factors for single-replicate enrichment factors. Hypothesis testing of feature distributions in enriched vs depleted fragments was carried out using non-parametric two-sided Mann–Whitney-Wilcoxon tests. For endogenous secretory protein enrichment, we used a Fisher's One-Sided Exact test. In case of more than 10 comparisons the significance of *p*-values was corrected using Benjamini–Hochberg multiple hypotheses testing. All analyses were performed using the R programming language (www.R-project.org), except for the correlation calculations of flow cytometric median fluorescence intensity vs enrichment factors of single clones, qPCR normalization efficiency comparisons, and *P. pastoris* growth, which were calculated using GraphPad Prism v7 and v9.

**Datasets for binary classification.** Two machine learning approaches were explored to investigate to what extent secretable and non-secretable fragments can be distinguished by primary sequence: a gradient boosted decision tree[47] model, and a convolutional neural network model[48]. Both approaches were constructed to perform this binary classification task, and were trained and evaluated on the same datasets.

The *S. cerevisiae* and *P. pastoris* datasets contain 148,156 and 151,761 protein fragments respectively, of which the (non-)secretability was consistent across the three replicates of the sorting experiment. In *S. cerevisiae* a total of 11,625 fragments were found to be consistently secreted (or enriched), and 136,531 fragments were found to be consistently non-secreted (or depleted). For *P. pastoris*, 10,404 secreted and 141,357 non-secreted fragments were found. Furthermore, we only retained fragments with a sequence length of at least 50 amino acids in the dataset, as we consider shorter sequences irrelevant because they do not fold properly. This resulted in the final dataset properties as shown in Supplementary Table 14.

Due to the imbalance between positive and negative samples in the dataset, the performance of the models was evaluated using the area under the curve of the receiver operating characteristic (AUROC) metric, as it is relatively insensitive to changes in class distribution. Instead of working with a fixed class probability threshold, the AUROC takes the ratio of detected enriched fragments (true positive rate) against the ratio of correctly assigned depleted fragments (false positive rate) for all possible thresholds. The AUC of this curve determines the performance, where a value of 1.0 indicates the best achievable performance and random prediction achieves a value of 0.5. The AUROC can also be seen as the probability that a randomly sampled enriched fragment has a higher predicted value than a randomly sampled depleted fragment.

A 10-fold cross-validation (CV) scheme was deployed to calculate the performance over the full dataset. To avoid bias between training and test data during this CV, folds were constructed in a way that all fragments originating from one gene belong to the same fold. If this measure would be disregarded, correct predictions on test data might be a result of sequence similarity and the model overfitting on training data, resulting in overly optimistic results. Simultaneously, folds were constructed to maintain similar class distributions.

The restrictive data selection scheme, requiring consistency over three replicates, resulted in a multitude of unused fragments. Therefore, in addition to a cross-validation over the full dataset, extra datasets were composed to further

validate the prediction models and the multi-replicate setup. For both *S. cerevisiae* and *P. pastoris*, fragments that were consistently enriched or depleted in solely two of the three replicates were selected, and divided over two separate datasets depending on whether the third replicate yielded an enrichment between −1 and 1 ("Set A", Sc_2consistent_1uncertain and Pp_2consistent_1uncertain), or whether it was opposite to the consistent replicates ("Set B", Sc_2consistent_1opposite and Pp_2consistent_1opposite). An overview of the number of fragments extracted using this selection procedure is shown in Supplementary Table 15. As the distribution between positively and negatively labeled fragments differs from the cross-validation data, results were again quantified using the AUROC, with its insensitivity to class distributions allowing for a comparison between the resulting scores across datasets.

**Gradient boosting**. The dataset consists of protein fragments of variable length. Traditional machine learning techniques typically rely on equal sized feature vectors and do not support variable input sizes. To overcome this problem, we extracted feature vectors from primary sequence to ensure a fixed size of the feature vector.

Multiple physicochemical properties were considered when extracting the feature vectors. For each property, the data extraction was performed and a separate model was trained. Amino acid scales were collected for the following properties: polarity[94], hydrophobicity[95], average area buried[96], buried residues[97], bulkiness[98], molar refraction[99], recognition factors[100], molecular weight, transmembrane tendency[101], and peptide retention time on HPLC[102].

For each property, five groups of features are extracted, resulting in a vector of 40 values for each data sample:
- Relative amino acid frequency (20 features, independent of the property).
- Sequence length (1 feature, independent of the property).
- The values of the property for the first six (at the N-terminus) and the last six (at the C-terminus) amino acids (12 features).
- The average value of the property over the entire sequence (1 feature).
- The average value of that property per region, when dividing each fragment into six equal-length regions (6 features). Shorter sequences will have shorter regions.

We then built a gradient boosting classifier that takes these feature sets as input. One classifier takes the 40 features per protein fragment as input, and produces a probability for the secretability of that fragment. After training a classifier for each property, an ensemble model is constructed, taking the ten probabilities of the individual classifiers as input for a new gradient boosting classifier, which then produces a final probability.

The hyperparameters for the gradient boosting decision trees were determined for each fold of the cross-validation individually, using a randomized search. For this search, the data from the training set in this fold was used. The results for the gradient boosting classifiers are listed in Supplementary Table 16.

**Convolutional neural network**. In recent times, deep learning techniques have been widely adopted in proteomics[103–105]. Especially convolutional neural networks (CNN) have been successfully applied in this context, given their ability to be trained end-to-end from primary sequence (preventing the need for manual feature engineering), their ability to learn spatial relations independent of position, and their intuitive way of encoding sequence motifs in the first convolutional layer.

A potential hindrance of the typical CNN architecture is that it expects a fixed-size input to produce a fixed-size output. Given the variability of sequence lengths in the secretability datasets, we explored four strategies to deal with this variation. After a one-hot encoding and three blocks, each consisting of a convolutional layer, rectified linear unit (ReLU) activation function, dropout layer and max pooling layer, the output of the last block is transformed using one of the following methods:
- Global max pooling, being a max pooling operation over the full sequence.
- K-max pooling[106], where the K highest activations are kept (in their respective order) per channel.
- A bidirectional gated recurrent unit (GRU), where the last hidden states of each direction are concatenated.

As a baseline, we also pad the input sequence with zeros until a fixed length is reached, and truncate any proteins that go beyond this length. After doing this, no transformation to a fixed length is necessary anymore. We choose a maximum length of 200 amino acids, as this covers 99.8% of all considered fragments.

Finally, this fixed-size output is followed by a fully-connected layer, which is then connected to an output layer with a single neuron. A sigmoid is used to generate probabilities from the final activation. The final hyperparameters of the architecture were determined using a grid search, and are given in Supplementary Table 17. The results for each architecture are given in Supplementary Table 18.

**Identifying decisive input features**. A challenge for neural networks, and various other machine learning techniques for that matter, is their lack of inherent interpretability. Attribution methods have been developed to combat this issue. Here, we use the integrated gradients[107] method, which is based on the back propagation algorithm. The principle of backpropagation-based attribution methods is to first do a forward pass through the network, generating an output signal, and to then backpropagate that signal back to the input to see which parts of the input

sequence were responsible for that prediction. This yields a so-called attribution (or saliency) map, with a positive or negative contribution per amino acid toward the predicted secretability of the fragment. The magnitude of the contribution indicates how strongly it directs the network toward secretable (positive contribution) or non-secretable (negative contribution) prediction. The overall magnitude of contributions scales with the confidence of the model.

For each protein fragment in the test set of a given fold, we calculated the attribution map for the optimal model (with global max pooling). To investigate the general behavior of the model, we then aggregated them using two strategies:
- We calculated the average contribution per amino acid, regardless of where in the sequence it occurred.
- We divided each sequence into twenty regions, and calculated the average contribution per amino acid per region. This means that the first region contains the average contribution of amino acids that occurred in the first 5% of their respective sequences, the second region from 5 to 10%, etc.

**Reporting summary**. Further information on research design is available in the Nature Research Reporting Summary linked to this article.

## Data availability

Unprocessed fastq files of both screens have been deposited in the Sequence Read Archive (SRA) under BioProject accession code PRJNA357179. Lists of all detected, enriched or depleted fragments are available on Figshare (http://figshare.com/s/82bb61370d7024f6fb09 for *S. cerevisiae* screens and http://figshare.com/s/cace104b0ffc5a57811f for *P. pastoris* screens), as are the lists of Pfam hits for all representative fragments (http://figshare.com/s/052370ec40154c09fb68), and CATH/Gene3D hits for those fragments mapping to PDB structures (http://figshare.com/s/5a8ca88d27168243c9fe). The data has been integrated in a web interface, available at http://iomics.ugent.be/secrify/search, for easy browsing by biologists interested in secretability of fragments of particular proteins of interest (Supplementary Fig. 10). These fragments are visually mapped to the PDB model of the protein's structure, where such structure is available. Source data are provided with this paper.

## Code availability

Code used and primary analysis of SECRiFY data can for sequencing data processing is available through Figshare (https://doi.org/10.6084/m9.figshare.5349979). The code to generate the Pfam hits and PBD mapping can be accessed via Github (https://github.com/Pathmanaban/SECRiFY_PDB_processing, https://doi.org/10.5281/zenodo.5542734), as can the code for training, evaluating and visualizing a convolutional neural network for secretability prediction (https://github.com/jasperzuallaert/SecrifyDL, https://doi.org/10.5281/zenodo.5541041), as well as the code for the gradient boosted decision tree modeling (https://github.com/RobbinBouwmeester/SECRiFY_xgb, https://doi.org/10.5281/zenodo.5541418).

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

## Acknowledgements

The authors thank the VIB Nucleomics Core for Illumina sequencing, as well as M. Vuylsteke for statistical insights, K. Vandewalle for help with western blot sample preparation, and Y. Dondelinger for help with plasmid construction. We thank Lennart Martens for freeing up time to discuss the project and to allow D.M., D.T., E.V. and P.R. to spend time on this project away from their main tasks. This work was supported by a Ghent University BOF PhD Fellowship (M.B. and H.E.), a PhD Fellowship from the Research Foundation Flanders (FWO) (M.B. and H.E.), an FWO research grant (G.0276.13N) (N.C.) and an ERC Consolidator grant no. 616966 (N.C.). E.V. is a postdoctoral research fellow of the Research Foundation Flanders. P.R. and W.V. acknowledge support from the Research Foundation Flanders through grant number G.0328.16N. W.D. and J.Z. are funded by the Ghent University Global Campus.

## Author contributions

M.B. performed library constructions, methods development and optimization, screenings, sequencing data processing and analyses, and wrote the manuscript. P.R. performed structure-based data-interpretation under the supervision of E.V. and W.V., W.V. ran, analyzed and visualized the biophysical predictions. S.D. and R.B. developed the gradient boosting classifiers, W.D. and J.Z. developed the deep learning classifiers, and together they analyzed the prediction results and feature contributions. D.M. and D.T. developed the website interface and the underlying database. B.V.M. and H.E. provided additional experimental support. N.C. conceived the project, and assisted in experimental design, interpretation, and manuscript writing. All authors read, revised, and approved the final version of the manuscript.

## Competing interests

The authors declare no competing interests.
