## [Peer Review File · Nature Communications]

Reviewers' Comments:

Reviewer #1:

Remarks to the Author:

> What are the major claims of the paper?

Boone et al. present an extensive study where they describe a new method, SECRiFY, to systematically assess the secretion of domain-sized recombinant protein fragments at proteome scale using the secretory systems of two yeasts, *S. cerevisiae* and *P. pastoris*.

The technology combines yeast surface display with an efficient method for constructing normalized cDNA fragment libraries with assessment of the screen output using massively parallel sequencing. It goes far beyond the structural genomics-style, clone-by-clone analyses where defined constructs are tested, and moves into the territory of random libraries with assessment of what emerges when presented to a selective system, here yeast secretion.

Through analysis of the unsecreted vs secreted constructs, the authors have published an online database (<https://iomics.ugent.be/secrify/>) permitting researchers to assess how their proteins of interest behaved when processed through SECRiFY, with alignments and mapping onto PDB protein structures. They then used advanced machine learning methods to deduce secretability characteristics that can then be used in a predictive manner i.e. that fragments of a define target could be assessed for secretability and potentially optimized in silico.

> Are they novel and will they be of interest to others in the community and the wider field?

The SECRiFY technology has similarities with other methods attempting to identify stable polypeptides from random cDNA libraries e.g. the selectively infective phage (SIP) technology of Pluckthun et al., but differs in that protein stability per se is not the phenotype being studied, rather secretability in either of two yeast strains, a holistic phenotype that encompasses expression, folding (or not), protease resistance, passage through the yeast secretory system and stable surface display.

I have the impression from the manuscript, given the size selection range and the human targetted libraries, plus first paragraph of Results (page 3) that this work may have originated as a domain-trapping approach. This would be of immense general interest for identifying domains of novel fold, or those unpredictable from their sequence, for downstream studies in functional proteomics or structural biology. The results however, indicate that domains are not identified using SECRiFY, even selected against by the constraints of the system. What is described is a system that permits identification of recombinant polypeptide fragments that are successfully displayed on yeast. The general interest of this is less clear than for a domain-trapping tool since the practical applications are less obvious and the biological relevance of the fragments identified is questionable. Nevertheless, the data have been used for a more fundamental study into the properties of polypeptides compatible with yeast secretion, an important process industrially and where conclusions may be applicable more generally to protein secretion in other organisms (though this would require significant further study to confirm).

> Originality of the conclusions.

Based on the results, the paper makes a number of conclusions:

- The library construction that is ORF directional and normalized is exceptional and ingenious. The subsequent screen by yeast surface display and cell sorting seems very effective.
- As with any library selection, the system has limitations and is not unbiased. The output was, disappointingly, lacking in stable structured domains, limiting SECRiFY's applicability for structural and functional studies, and de novo protein design (line 391). There was no enrichment of naturally secreted proteins either, maybe due to the human nature of the targets, or the constraints of the screen. These observations are well described and discussed in the manuscript.
- A major effort was pursued to derive meaning from the experimental data and this is interesting, but it is not clear if this informs the reader of anything beyond the compatibility of (human)

protein fragments with this particular screen.

- Successfully secreted polypeptides were often predicted to lack structure or have minimal secondary structure content. I am not convinced that mapping fragments on to PDB structures and making conclusions about folds (line 330) is valid since these folds may well not be present in shorter fragments that lack some of the structure due to the absence of proper hydrophobic core and resulting structural stabilization. Similarly, the EDIL3 enriched sub-fragment (Fig. 3e) is shown structured, but whether it actually is structured is questionable and not demonstrated experimentally – perhaps this fragment it is just smaller and so secreted better?
- There is some sequence bias, notably acidic residues (Fig. 4c), possibly Pro and Cys also. Positively charged aa are under-represented. This is discussed on page 13, with a possible artefactual situation being proposed as the reason (antibody staining and accessibility due to phosphomannose protein charge).
- Similarly, the nature of the display construct (N-ter secretion peptide, C-ter Sag1) is proposed as another possibility for bias of the experimental output.

> Is the work convincing, and if not, what further evidence would be required to strengthen the conclusions?

The experimental part of the work is really convincing – the library construction and screen use state-of-the-art methods (library sorting, next gen sequencing, machine learning).

The authors cite library approaches for identifying soluble domain-based constructs. These work because of saturation-level testing of domain boundaries, so that rare clones whose folding and soluble expression are sensitive to exact N- or C-ter ends are tested (line 109). The fact that SECRiFY makes quite big libraries, but which are focused on a mixed pool of thousands of different targets means that any one target protein is sampled very sparsely. It would be useful to see data, that should exist from the NGS of the naïve library, for several defined targets from the library showing the depth of sampling and position of termini. How many fragments per protein are being tested? Are these fragments positionally biased or is it like an unbiased sliding window of fragments with all internal fragments within the defined size range are represented equally?

An output of selected secreted clones is obtained and an attempt to rationalize this is made. However, it is not clear what is really being measured here – is it only the output of this specific screening system with all its constraints? Figure S4 shows also that negative clones from the screen are still secretable i.e. that there is a non-negligible level of false negatives. So, how generally valid are the conclusions being drawn? Can these be extended to secretion in other cellular systems, either yeast with different vectors, or other eucaryotes? The output, easily accessible via the well-constructed database and web server, would seem to have little practical utility e.g. structural studies (line 380). Submitting some well-known targets to the web server reveals secreted fragments that don't seem to be biologically relevant to function or structure. So what is their relevance beyond compatibility with the vectors and screening approach of SECRiFY?

> On a more subjective note, do you feel that the paper will influence thinking in the field?

At a technical level, the protocols for constructing directional normalized cDNA fragment libraries will be a useful resource. I find it hard to say if the analysis and conclusions inform us on the rules that underpin general protein secretion, rather than those specific to this system.

> Please feel free to raise any further questions and concerns about the paper.

The supplementary info is thorough, but very long. With 38 pages of supplementary info containing 18 supp figures and 17 tables of supp data, it is difficult to digest.

> We would also be grateful if you could comment on the appropriateness and validity of any statistical analysis, as well the ability of a researcher to reproduce the work, given the level of detail provided.

- The online methods are very well written.

- The figures are well drawn and clear.
- The manuscript is well written in a clear logical style.
- The basic statistics seem fine – others are beyond my experience, as are the finer details of the machine learning.

> Minor comments

- The SECRiFY webpage refers to a non-existent Boone et al. publication in 2017 with non functional DOI.
- Online methods – ensure uniform spacing between numbers and their units (1 h vs 1h)

Reviewer #2:

Remarks to the Author:

In the manuscript entitled

“Massively parallel interrogation of protein fragment secretability using SECRiFY reveals features influencing secretory system transit”

The authors have developed a new method named “SECRiFY (SECRetability screening of Recombinant Fragments in Yeast)” to monitor protein secretion of human gene fragments from yeast and to define protein secretability predictors.

The authors show that protein secretability is a learnable feature of protein sequences with claimed significance for heterologous protein expression and de novo protein design.

The described work is convincing, within its limitations see below, and the findings that secondary structure patterns differ between secreted and non-secreted fragments are novel and interesting. Also, the machine learning predictive algorithm built is novel and interesting.

Methods are comprehensively presented and since the experiments have been performed in two different organisms in triplicate the reviewer is confident that results can be repeated. *S. cerevisiae* (Baker’s yeast) is not a go to organism for protein secretion and thus it makes sense to complement the data with an additional data set using *Pichia pastoris*.

Major concerns

I find it difficult to agree with the impact of the method as described for the field of heterologous (recombinant) protein expression.

Unpredictability of heterologous protein secretion (page 2 line 53)” and “Despite the tremendous strides made in the field of recombinant protein production, heterologous expression remains highly unpredictable (page 11 line 357)”

The reasons for this are the following:

1) Experimental setup

a) The authors have chosen two similar expression systems (*S. cerevisiae* and *P. pastoris*). Neither are key organisms within the field of recombinant protein expression of secreted proteins, IMHO, with *P. pastoris* being a niche system for production of membrane proteins.

Significant impact would have been added for the field of recombinant protein production if a mammalian host could have been explored. Similar to what was done in

- Construction and Screening of a Lentiviral Secretome Library

o <https://doi.org/10.1016/j.chembiol.2017.05.017>

- The screening setup is similar with positive selection of secreted FL clones from a human ORFeome library

On page 5 line 149 it is stated “we further reasoned that surface display could be used as a proxy for productive secretion” This technique was used in the above reference and should thus be cited.

b) The size of fragments screened varied between 50-100 amino acids. Thus, information about secretability of proteins are not provided by the authors.

- The manuscript would become much stronger if small full-lengths secreted proteins of a certain size (eg. < 20 kDa) could have been included. Could this type of data be included for a small subset of reference proteins?

c) The endogenous signal peptide, normally found at the N-terminus of the coding sequence, was not included in experimental setup and is not discussed.

This discussion needs to be included

<https://doi.org/10.1016/j.tibs.2006.08.004>

Also, the limitations of using the alpha mating factor prepro sequence (page 13 line 434) should be outlined

Also, it would be valuable to assess whether fragments corresponding to the first domain after the native signal peptide are enriched in the selected fractions (Table 1 and page 11 line 351). Did the authors please check for this?

2) The description of the current status of the field is not up to date IMHO

- In the introductory section (page 2 lines 57-68) the description of the literature covers more than 10 year old efforts from mostly Structural Genomics Consortia Initiatives, with *E. coli* mostly used as expression host in these initiatives. Considerable advances have been made in the use of heterologous expression hosts and as a result success rates have increased considerably. Also, synthetic biology has revolutionized the field with cell-free expression becoming more and more attractive for expression screening of protein domains and production of large quantities of protein. This cell-free process will no doubt be quicker and more stream-lined than the process outlined in this manuscript.

- It is not correct as stated by authors (page 2 line 58) that success rates for purified protein lies between 20-50%.

1. Discovery of a cytokine and its receptor by functional screening of the extracellular proteome. Here, success rates for secreted protein production were reported to be ~70%

DOI: 10.1126/science.1154370

2. Expression of the human secretome in mammalian cells. This articles describes the expression of all FL secreted proteins including detailed analysis for failure

<https://doi.org/10.1016/j.nbt.2020.05.002>. Success rate for purified protein is 60%

3) Discussion of results

1) The authors show no positive correlation between secreted fragments and prediction to be secreted by the presence of a signal peptide.

Page 11 line 42: "Protein secretability does not correlate with endogenous secretion"

The reviewer find this surprising and the discussion around this is weak. For example page 11 line 352 – which features are the authors thinking of that would determine processing of proteins via the secretory pathway?

Page 2 line 43 "what sequence or structural features control ER export kinetics". This sentence is a bit odd as it stand since there is no mention of the well-known KDEL ER retention signal

2) Did the authors consider validating their findings by assessing translatability of fragments in a cell-free system?

For example 20 fragments that were enriched and 20 fragments that were depleted could be tested in a Wheat Germ lysate/HeLA cell lysate and translational efficiency could be measured with removing the complexity and artefacts of secretion via the *S. cerevisiae*/*P. pastoris* system

3) Page 13 line 433: The authors share their concern for the surface display method used and tethering of the secreted protein fragment to the yeast cell wall. Did the authors consider using secretion of an enzyme as a positive control (eg alkaline phosphatase) and to measure enzymatic activity in the conditioned medium?

Minor comments

Why where different sources of human RNA used for generation of the libraries in *S. cerevisiae* and *P. pastoris*? Why was a mixture of RNA from different sources used for the latter experiments? This is not at all discussed.

Fragments were optimized to be around 500 bp. What was the distribution size of fragments? Even though median size of protein domains are 50-100 there are a considerable number of domains that contain additional amino acids. Also, tertiary folds involve interaction of amino acids in different domains. This is not at all discussed.

Reviewer #3:

Remarks to the Author:

Recombinant production of proteins remains a largely trial-and-error process. To understand the sequence/structural-determinants of recombinant expression, the authors used secretion in yeast as a proxy. The authors constructed a domain-sized library from the human proteome using some clever tricks, then used yeast surface display + deep sequencing to identify protein fragments that can travel through the yeast secretory system to be displayed on the cell surface. This is referred to as "secretability". Based on training a ML model on their experimental enrichment/depletion data, the authors find that more flexible and intrinsically disordered proteins are more secretable, and secretory proteins are NOT more secretable.

While the study is technically nicely done, the patterns in results are somewhat interesting and shed light on secretion machinery in yeast, my two major concerns are that (1) these are randomly generated fragments, and the applicability to recombinant expression of full-length (typically multi-domain) proteins is not clear, and (2) surface display may or may not necessarily mean a properly folded protein (for folded proteins, these could be molten globules). A minor concern is that while ML is given prominence in the paper, it is not clear what new insights were gained that were already not obtained from biophysical predictions. If there were new insights, they were not experimentally tested. Could the authors potentially show utility of their method by improving the expression of some fragment using the rules they have discovered?

Manuscript NCOMMS-20-47011

“Massively parallel interrogation of protein fragment secretability using SECRiFY reveals features influencing secretory system transit” by Boone et al.

Response to reviewers

Dear reviewers,

we sincerely appreciate the time you have taken to carefully, critically, and thoroughly read our manuscript. We carefully considered and addressed most of the issues you have raised. Please accept our responses (in *red italics*) to your questions and comments below.

Referee #1

1. Major claims

Boone et al. present an extensive study where they describe a new method, SECRiFY, to systematically assess the secretion of domain-sized recombinant protein fragments at proteome scale using the secretory systems of two yeasts, *S. cerevisiae* and *P. pastoris*.

The technology combines yeast surface display with an efficient method for constructing normalized cDNA fragment libraries with assessment of the screen output using massively parallel sequencing. It goes far beyond the structural genomics-style, clone-by-clone analyses where defined constructs are tested, and moves into the territory of random libraries with assessment of what emerges when presented to a selective system, here yeast secretion.

Through analysis of the unsecreted vs secreted constructs, the authors have published an online database (<https://iomics.ugent.be/secrifly/>) permitting researchers to assess how their proteins of interest behaved when processed through SECRiFY, with alignments and mapping onto PDB protein structures. They then used advanced machine learning methods to deduce secretability characteristics that can then be used in a predictive manner i.e. that fragments of a define target could be assessed for secretability and potentially optimized in silico.

2. Novelty and interest to community or wider field

The SECRiFY technology has similarities with other methods attempting to identify stable polypeptides from random cDNA libraries e.g. the selectively infective phage (SIP) technology of Pluckthun et al., but differs in that protein stability per se is not the phenotype being studied, rather secretability in either of two yeast strains, a holistic phenotype that encompasses expression, folding (or not), protease resistance, passage through the yeast secretory system and stable surface display.

I have the impression from the manuscript, given the size selection range and the human targetted libraries, plus first paragraph of Results (page 3) that this work may have originated as a domain-trapping approach. This would be of immense general interest for identifying domains of novel fold, or those unpredictable from their sequence, for downstream studies in functional proteomics or

structural biology. The results however, indicate that domains are not identified using SECRiFY, even selected against by the constraints of the system. What is described is a system that permits identification of recombinant polypeptide fragments that are successfully displayed on yeast. The general interest of this is less clear than for a domain-trapping tool since the practical applications are less obvious and the biological relevance of the fragments identified is questionable. Nevertheless, the data have been used for a more fundamental study into the properties of polypeptides compatible with yeast secretion, an important process industrially and where conclusions may be applicable more generally to protein secretion in other organisms (though this would require significant further study to confirm).

Response: While we are tackling to demonstrate the direct applicability of this tool in further studies, we de-emphasized the practical utility of SECRiFY in the revised version of this manuscript, in line with similar comments from Reviewer #3.

3. Originality of the conclusions

Based on the results, the paper makes a number of conclusions:

- The library construction that is ORF directional and normalized is exceptional and ingenious. The subsequent screen by yeast surface display and cell sorting seems very effective.
- As with any library selection, the system has limitations and is not unbiased. The output was, disappointingly, lacking in stable structured domains, limiting SECRiFY's applicability for structural and functional studies, and *de novo* protein design (line 391). There was no enrichment of naturally secreted proteins either, maybe due to the human nature of the targets, or the constraints of the screen. These observations are well described and discussed in the manuscript.
- A major effort was pursued to derive meaning from the experimental data and this is interesting, but it is not clear if this informs the reader of anything beyond the compatibility of (human) protein fragments with this particular screen.

Response: The major insight obtained from our work is that successful secretion of a foreign protein fragment is not random and relies on patterns intrinsic to that polypeptide's sequence. While more work will be required to determine generality across other proteomes, or which biosynthesis or quality control proteins these specific sequence features depend on, this study substantiates the potential for such findings.

- Successfully secreted polypeptides were often predicted to lack structure or have minimal secondary structure content. I am not convinced that mapping fragments on to PDB structures and making conclusions about folds (line 330) is valid since these folds may well not be present in shorter fragments that lack some of the structure due to the absence of proper hydrophobic core and resulting structural stabilization. Similarly, the EDIL3 enriched sub-fragment (Fig. 3e) is shown structured, but whether it actually is structured is questionable and not demonstrated experimentally – perhaps this fragment it is just smaller and so secreted better?

Response: *It is indeed unknown whether all secretable polypeptide fragments that map to PDB structures actually do form well-folded structures reminiscent of their structure in the full-length protein. Since we lack suitable methods to efficiently determine the structure of thousands of polypeptides at the same time, experimental validation of even a subset of fragments is laborious. We therefore do not claim that SECRiFY selects for such folded polypeptides, but emphasize throughout the manuscript that it enriches for fragments that either successfully pass through the cell's quality control systems or successfully evade them. That being said, while we could not detect a global difference in the predicted propensity to fold, and despite the absence of structural data of the displayed polypeptides, it is striking that we can pick up significant differences in the potential occurrence of specific domain folds and architectures. This is hinting at a meaningful role for conformation and fold in secretability. To avoid any confusion, we adapted the phrasing of this paragraph to reflect this more clearly, and highlight this again in the Discussion section.*

- There is some sequence bias, notably acidic residues (Fig. 4c), possibly Pro and Cys also. Positively charged aa are under-represented. This is discussed on page 13, with a possible artefactual situation being proposed as the reason (antibody staining and accessibility due to phosphomannose protein charge).
- Similarly, the nature of the display construct (N-ter secretion peptide, C-ter Sag1) is proposed as another possibility for bias of the experimental output.

4. Is the work convincing, and if not, what further evidence would be required to strengthen the conclusions?

The experimental part of the work is really convincing – the library construction and screen use state-of-the-art methods (library sorting, next gen sequencing, machine learning).

The authors cite library approaches for identifying soluble domain-based constructs. These work because of saturation-level testing of domain boundaries, so that rare clones whose folding and soluble expression are sensitive to exact N- or C-ter ends are tested (line 109). The fact that SECRiFY makes quite big libraries, but which are focused on a mixed pool of thousands of different targets means that any one target protein is sampled very sparsely. It would be useful to see data, that should exist from the NGS of the naïve library, for several defined targets from the library showing the depth of sampling and position of termini. How many fragments per protein are being tested? Are these fragments positionally biased or is it like an unbiased sliding window of fragments with all internal fragments within the defined size range are represented equally?

Response: *Our library synthesis method using random primed normalized libraries was designed to minimize positional bias. The fragment position across proteins in the naive library (Figure R1 a,d) attests to this and is reminiscent of the fragment coverage distribution for random primed nucleic acid libraries, with slightly lower coverage at protein termini, but this is less relevant given our focus on domain-sized fragment rather than full-length protein secretability. The depth of sampling mostly depends on library normalization efficiency and number of transformants, which*

was lower for the *S. cerevisiae* proof-of-concept library (covering 26% of the canonical human proteome with at least 3 reads) than for the *P. pastoris* library (38% of the proteome). While the majority of proteins with at least one fragment are sampled with <10 fragments, and the libraries were thus indeed not saturating, our datasets are still the largest of this type and allowed for in-depth unbiased pattern discovery.

Figure R1. Protein coverage statistics of SECRiFY library fragments in naive libraries. (a,d) Fragment termini position on a universal protein model of 100 amino acids for the *S. cerevisiae* and *P. pastoris* libraries, resp. (b, e) Binned number of unique fragments per protein with at least one in-frame fragment for the *S. cerevisiae* and *P. pastoris* libraries, resp. (c,f) Unique fragment number distribution for all proteins with at least one in-frame fragment for the *S. cerevisiae* and *P. pastoris* libraries, resp.

An output of selected secreted clones is obtained and an attempt to rationalize this is made. However, it is not clear what is really being measured here – is it only the output of this specific screening system with all its constraints? Figure S4 shows also that negative clones from the screen are still secretable i.e. that there is a non-negligible level of false negatives. So, how generally valid are the conclusions being drawn? Can these be extended to secretion in other cellular systems, either yeast with different vectors, or other eucaryotes?

Response: As we mention in pt. 3, the major insight obtained from our work is that successful secretion of a foreign protein fragment is not random and relies on patterns intrinsic to that polypeptide's sequence. The lower negative predictive value means that interpretation of features

affecting secretion must focus on those that affect secretability, and not non-secretability. As there are ± 15 times as many fragments in the depleted set, this relatively low negative predictive value was not an impediment for the machine learning methods to extract relevant information.

It is so far unknown to what extent the experimental specifics (eg different vector contexts, mammalian hosts, different proteome source etc) influence the details of these patterns, but the observation that the features we extract are very similar for two very different yeast species suggests a certain level of generality and highlights the robustness of the method. Our future work will focus on determining how universal these are across various sequence contexts and hosts.

The output, easily accessible via the well-constructed database and web server, would seem to have little practical utility e.g. structural studies (line 380). Submitting some well-known targets to the web server reveals secreted fragments that don't seem to be biologically relevant to function or structure. So what is their relevance beyond compatibility with the vectors and screening approach of SECRiFY?

Response: *as mentioned above, our future work aims to demonstrate the direct applicability of this tool in further studies. To address this concern in the current manuscript, however, we decided to de-emphasized the practical utility of SECRiFY in the revised version of the main text. The web server will remain accessible but we now only refer to it as a resource for interested readers in the data availability section of the Online Methods.*

5. On a more subjective note, do you feel that the paper will influence thinking in the field?

At a technical level, the protocols for constructing directional normalized cDNA fragment libraries will be a useful resource. I find it hard to say if the analysis and conclusions inform us on the rules that underpin general protein secretion, rather than those specific to this system.

Response: *We refer to our responses to point 3 and 4 above for this comment.*

6. Please feel free to raise any further questions and concerns about the paper.

The supplementary info is thorough, but very long. With 38 pages of supplementary info containing 18 supp figures and 17 tables of supp data, it is difficult to digest.

Response: *Our datasets are large and very information-rich, but we distilled the most important observations for the main text and figures, while keeping all the rest readily accessible for the readers in the supplementary. Of note, figures may look duplicated but we did the same analysis for screens in two different organisms.*

7. We would also be grateful if you could comment on the appropriateness and validity of any statistical analysis, as well the ability of a researcher to reproduce the work, given the level of detail provided.

- The online methods are very well written.
- The figures are well drawn and clear.
- The manuscript is well written in a clear logical style.

- The basic statistics seem fine – others are beyond my experience, as are the finer details of the machine learning.

8. Minor comments

- The SECRiFY webpage refers to a non-existent Boone et al. publication in 2017 with non functional DOI.

Response: Thank you for pointing this out – this referred to an earlier version of this manuscript and has now been updated.

- Online methods – ensure uniform spacing between numbers and their units (1 h vs 1h)

Response: Corrected in the revised version of the manuscript.

Referee #2

In the manuscript entitled “Massively parallel interrogation of protein fragment secretability using SECRiFY reveals features influencing secretory system transit” The authors have developed a new method named “SECRiFY (SECretability screening of Recombinant Fragments in Yeast)” to monitor protein secretion of human gene fragments from yeast and to define protein secretability predictors. The authors show that protein secretability is a learnable feature of protein sequences with claimed significance for heterologous protein expression and de novo protein design. The described work is convincing, within its limitations see below, and the findings that secondary structure patterns differ between secreted and non-secreted fragments are novel and interesting. Also, the machine learning predictive algorithm built is novel and interesting. Methods are comprehensively presented and since the experiments have been performed in two different organisms in triplicate the reviewer is confident that results can be repeated. *S. cerevisiae* (Baker’s yeast) is not a go to organism for protein secretion and thus it makes sense to complement the data with an additional data set using *Pichia pastoris*.

Major concerns

I find it difficult to agree with the impact of the method as described for the field of heterologous (recombinant) protein expression. Unpredictability of heterologous protein secretion (page 2 line 53)” and “Despite the tremendous strides made in the field of recombinant protein production, heterologous expression remains highly unpredictable (page 11 line 357)”. The reasons for this are the following:

1) Experimental setup

- a) The authors have chosen two similar expression systems (*S. cerevisiae* and *P. pastoris*). Neither are key organisms within the field of recombinant protein expression of secreted proteins, IMHO, with *P. pastoris* being a niche system for production of membrane proteins. Significant impact would have been added for the field of recombinant protein production if a mammalian host could have been explored. Similar to what was done in Construction and Screening of a Lentiviral Secretome Library <https://doi.org/10.1016/j.chembiol.2017.05.017>.

The screening setup is similar with positive selection of secreted FL clones from a human ORFeome library. On page 5 line 149 it is stated “we further reasoned that surface display could be used as a proxy for productive secretion” This technique was used in the above reference and should thus be cited.

Response: *Although the share of recombinant proteins produced in *P. pastoris* is not as big as for CHO or HEK cells, there are plenty of Pichia-produced soluble proteins on the market (Kalbitor, Insugen, Shanvac, several Nanobodies, Shanferon etc – see <https://pichia.com/science-center/commercialized-products/>), and it is definitely not niche (for example, Walsh, G. *Biopharmaceutical benchmarks 2014*. *Nat Biotechnol* 32, 992–1000 (2014). <https://doi.org/10.1038/nbt.3040>). As a starting point for studying the eukaryotic secretory system in a high throughput manner in a model organism, baker’s yeast has been the obvious organism or choice for many decades, and its easy genetics will prove very valuable to assess the impact of knock-outs or knock-ins on secretability. Extending this study to mammalian systems is indeed an exciting next step and will be incorporated in future work.*

The Lentiviral Secretome paper this reviewer refers to had not been published at the time of our first submission, but we thank them for pointing it out and have now appropriately cited it in our manuscript.

b) The size of fragments screened varied between 50-100 amino acids. Thus, information about secretability of proteins are not provided by the authors. The manuscript would become much stronger if small full-lengths secreted proteins of a certain size (eg. < 20 kDa) could have been included. Could this type of data be included for a small sub-set of reference proteins?

Response: *We intentionally designed our method to study the secretability of short, domain-sized protein fragments rather than full-length proteins. As we argue in our manuscript (first paragraph in the Results section), we opted for this reductionist approach to simplify the identification of sequence or structural patterns that may influence secretion. Our models were thus trained on protein fragments of a certain size, not full-length proteins. The fact that most full-length proteins are multi-domain polypeptides that need to assemble or coordinate domain folding adds several dimensions to the problem, and disentangling these complexities may require a different approach. Whether the patterns we found can be extrapolated to full-length proteins will require new screens and is beyond the scope of this study. Of note, we are currently exploring this area in follow-up projects.*

c) The endogenous signal peptide, normally found at the N-terminus of the coding sequence, was not included in experimental setup and is not discussed. This discussion needs to be included <https://doi.org/10.1016/j.tibs.2006.08.004>. Also, the limitations of using the alpha mating factor prepro sequence (page 13 line 434) should be outlined. Also, it would be

valuable to assess whether fragments corresponding to the first domain after the native signal peptide are enriched in the selected fractions (Table 1 and page 11 line 351). Did the authors please check for this?

Response: *Since our goal was to look for sequence features independent of the signal sequence, all fragments in our library are cloned behind the same signal sequence – the alpha mating factor pre (and pro) sequence. As mentioned above, whether the patterns that we found hold up across different contexts (including different signal sequences) is currently unknown and the subject of ongoing work in our lab. Note that we cover the proteome independent of the endogenous signal sequence, and given the ‘sliding window’ mapping of the fragments generated, the majority of fragments do not encompass endogenous signal sequences, which would also appear behind the alpha prepro signal sequence and therefore not be N-terminal. As mentioned in the results, even for fragments that map to proteins that are endogenously secreted, no enrichment for this subclass is seen with SECRIFY.*

2) The description of the current status of the field is not up to date IMHO - In the introductory section (page 2 lines 57-68) the description of the literature covers more than 10 year-old efforts from mostly Structural Genomics Consortia Initiatives, with *E. coli* mostly used as expression host in these initiatives. Considerable advances have been made in the use of heterologous expression hosts and as a result success rates have increased considerably. Also, synthetic biology has revolutionized the field with cell-free expression becoming more and more attractive for expression screening of protein domains and production of large quantities of protein. This cell-free process will no doubt be quicker and more stream-lined than the process outlined in this manuscript. It is not correct as stated by authors (page 2 line 58) that success rates for purified protein lies between 20-50%.

- a) Discovery of a cytokine and its receptor by functional screening of the extracellular proteome. Here, success rates for secreted protein production were reported to be ~70% DOI: [10.1126/science.1154370](https://doi.org/10.1126/science.1154370)
- b) Expression of the human secretome in mammalian cells. This articles describes the expression of all FL secreted proteins including detailed analysis for failure <https://doi.org/10.1016/j.nbt.2020.05.002>. Success rate for purified protein is 60%

Response: *We acknowledge that success rates can vary wildly across hosts and heterologous proteins, and this has indeed generally improved over the years since the Structural Genomics consortia. We omitted these statistics from the revised version of the manuscript. We would, however, like to note that most success rate statistics are artificially high because a preselection is made for the constructs that are most likely to express in a certain host that is evolutionarily close to the protein of origin. The broader point we make in this paragraph is that, given a particular host (eg yeast) and a random protein from any other source, our knowledge of the requirements for successful expression (and we here particularly focus on secretory expression) is limited and whether or not the protein will make it out successfully is still more a matter of trial and error rather than something we can currently consciously design. Getting success rates of 60-70% for secretion of secretory mammalian proteins in a mammalian host still underlines that – aside from the downstream processes of purification after*

harvesting – there is still much to learn about the intrinsic biological factors that affect efficient protein passage through the secretory system.

Cell-free expression is a valuable approach in its own right, but we here focused on developing a method that would help us understand what the fundamental requirements are for passage through the secretory system, which inherently requires a eukaryotic cell.

3) Discussion of results

- a) The authors show no positive correlation between secreted fragments and prediction to be secreted by the presence of a signal peptide. Page 11 line 42: “Protein secretability does not correlate with endogenous secretion”. The reviewer find this surprising and the discussion around this is weak. For example page 11 line 352 – which features are the authors thinking of that would determine processing of proteins via the secretory pathway? Page 2 line 43 “what sequence or structural features control ER export kinetics”. This sentence is a bit odd as it stands since there is no mention of the well-known KDEL ER retention signal.

Response: *As mentioned in the discussion, SECRiFY detects secretion at the fragment level, potentially causing some features affecting full-length protein secretion to be missed. Nonetheless, the absence of correlation also underlines that SECRiFY assesses secretability, i.e. the capacity to be secreted, rather than actual endogenous cellular localization to the secretory system. Indeed, just like most proteins are only marginally stable, endogenous secretory proteins evolved for function, not secretability.*

It is currently strikingly unknown what features – beyond simple KDEL retention signals, or N-glycan sites or disulfide bonds, for example – affect secretability, and uncovering which features these are is the exact purpose of a method like SECRiFY.

- b) Did the authors consider validating their findings by assessing translatability of fragments in a cell-free system? For example 20 fragments that were enriched and 20 fragments that were depleted could be tested in a Wheat Germ lysate/HeLA cell lysate and translational efficiency could be measured with removing the complexity and artefacts of secretion via the *S. cerevisiae*/P. *pastoris* system.

Response: *As mentioned above, since the goal was to study passage through the yeast secretory system, we do not consider testing these fragments in a cell-free system of relevance for this paper. These environments are very different and would not elevate our understanding of secretability.*

- c) Page 13 line 433: The authors share their concern for the surface display method used and tethering of the secreted protein fragment to the yeast cell wall. Did the authors consider using secretion of an enzyme as a positive control (eg alkaline phosphatase) and to measure enzymatic activity in the conditioned medium?

Response: *We would like to refer the reviewer to the body of work, from our and other labs, reporting successful yeast display of heterologous enzymes and other proteins without affecting activity or other protein function, and correlating display levels with protein secretion levels (for example, DOIs 10.1006/jmbi.1999.3130, 10.1038/77325, 10.1128/AEM.02427-06,*

10.1038/nbt785, 10.1016/j.jim.2011.10.003). This is mentioned in the discussion. In a similar vein, we've also shown that this relationship holds true with our plasmid vectors using point mutant variants of human lysozyme that differ in their propensity to fold in the ER (see figure R2 below).

Figure R2. Relationship between free secretion vs surface display levels of human lysozyme point mutants in *P. pastoris*. Lysozyme variants were secreted as FLAG-hLysozyme-V5 and measured by western blotting, or displayed as FLAG-hLysozyme-V5 followed by flow cytometry. FU= functional units normalized to WT lysozyme as 1. MFI= Median Fluorescence Units.

Minor comments

- Why were different sources of human RNA used for generation of the libraries in *S. cerevisiae* and *P. pastoris*? Why was a mixture of RNA from different sources used for the latter experiments? This is not at all discussed.

Response: We started with a small test library from HEK293 cells for *S. cerevisiae* as proof-of-concept. As mentioned in the main text (“selected to maximize the number of expressed human genes based on ENCODE transcriptome data”, line 212), since not all cells express all genes, and hence to cover a broader range of the human transcriptome, we calculated which cell lines diverged the most in transcriptome expression (using publicly available data from the ENCODE project) and used a combination of 4 available cell lines. This increased our coverage of the human transcriptome from 26% to 38%.

- Fragments were optimized to be around 500 bp. What was the distribution size of fragments? Even though median size of protein domains are 50-100 there are a considerable number of domains that contain additional amino acids. Also, tertiary folds involve interaction of amino acids in different domains. This is not at all discussed.

Response: We refer to Supplementary Figure 6 for all fragment distribution plots. Statistics on the extent of discontinuity in domains vary but have been reported to be around 15-20% depending on the algorithm (DOI 10.1371/journal.pcbi.0030232, DOI 10.1371/journal.pone.0141541). An analysis of the CATH structural domain database (v4.0.0, as of April 7th, 2016) calculates that 14.62% of all known domains are discontinuous, although this drops to 11.35% for the human proteome. Indeed, our random fragmentation and size selection approach does not guarantee coverage of all possible domains, but was optimized to encompass a large fraction of them.

Recombinant production of proteins remains a largely trial-and-error process. To understand the sequence/structural-determinants of recombinant expression, the authors used secretion in yeast as a proxy. The authors constructed a domain-sized library from the human proteome using some clever tricks, then used yeast surface display + deep sequencing to identify protein fragments that can travel through the yeast secretory system to be displayed on the cell surface. This is referred to as "secretability". Based on training a ML model on their experimental enrichment/depletion data, the authors find that more flexible and intrinsically disordered proteins are more secretable, and secretory proteins are NOT more secretable.

While the study is technically nicely done, the patterns in results are somewhat interesting and shed light on secretion machinery in yeast, my two major concerns are that

(1) these are randomly generated fragments, and the applicability to recombinant expression of full-length (typically multi-domain) proteins is not clear

Response: *We here refer to our response to question 1b from reviewer #2: we intentionally designed our method to study the secretability of short, domain-sized protein fragments rather than full-length proteins. As we argue in our manuscript (first paragraph in the Results section), we opted for this reductionist approach to simplify the identification of sequence or structural patterns that may influence secretion. Our models were thus trained on protein fragments of a certain size, not full-length proteins. The fact that most full-length proteins are multi-domain polypeptides that need to assemble or coordinate domain folding adds several dimensions to the problem, and disentangling these complexities may require a different approach. Whether the patterns we found can be extrapolated to full-length proteins will require new screens and is beyond the scope of this study. Of note, we are currently exploring this area in follow-up projects.*

(2) surface display may or may not necessarily mean a properly folded protein (for folded proteins, these could be molten globules).

Response: *indeed, and this is a striking observation given that the secretory system evolved to avoid secretion of misfolded proteins – this is why we have extensive processes like QC. As we note above in our response to point 3 of reviewer #1, it is indeed unknown whether all secretable polypeptide fragments that map to PDB structures actually do form well-folded structures reminiscent of their structure in the full-length protein. Since we lack suitable methods to efficiently determine the structure of thousands of polypeptides at the same time, experimental validation of even a subset of fragments is laborious. We do not claim that SECRiFY selects for such folded polypeptides, but emphasize throughout the manuscript that it enriches for fragments that either successfully pass through the cell's quality control systems or successfully evade them. That being said, while we could not detect a global difference in the predicted propensity to fold, and despite the absence of structural data of the displayed polypeptides, it is striking that we can pick up significant differences in the potential*

occurrence of specific domain folds and architectures. This is hinting at a meaningful role for conformation and fold in secretability. To avoid any confusion, we adapted the phrasing of this paragraph to reflect this more clearly, and highlight this again in the Discussion section.

A minor concern is that while ML is given prominence in the paper, it is not clear what new insights were gained that were already not obtained from biophysical predictions. If there were new insights, they were not experimentally tested. Could the authors potentially show utility of their method by improving the expression of some fragment using the rules they have discovered?

Response: *it is important to stress that while our models were able to learn patterns in the data, pattern finding does not immediately translate to reliable design, which is much harder to do since patterns generally only contain a fraction of the information necessary for design. We acknowledge that the use of the word 'predictability' might have cast some confusion, and have adjusted our language in the manuscript to clarify this and emphasize the more fundamental insights we obtained with our study. That being said, we now included a new analysis of experimental data that was not included in the first version of the manuscript, using fragments that were detected in only 2 of the 3 replicates of each screen. Our machine learning models were able to reach similar accuracy for this completely independent dataset, providing an alternative validation of the patterns observed on a much larger scale than had been possible for wet-lab validation of individual fragments.*

Reviewers' Comments:

Reviewer #1:

Remarks to the Author:

My main concerns regarding the claimed practical utility of the SECRiFY method to recombinant protein expression for structural biology have been addressed in the revised manuscript. Similarly, the structural significance of the protein fragments, derived by mapping them onto larger more complete PDB structures, has been de-emphasized. The revised version focuses more on the fundamental aspects of polypeptide secretability in yeast which, in itself, should be of interest to readers of the manuscript. My points have thus been addressed by the authors.

Reviewer #2:

Remarks to the Author:

The authors have reworked the manuscript considerably and the new version is very much clearer. However, there is in my opinion remaining some unclarity around scope and impact of work. See comments below.

Comment 2

The authors reply that

" We acknowledge that success rates can vary wildly across hosts and heterologous proteins, and this has indeed generally improved over the years since the Structural Genomics consortia. We omitted these statistics from the revised version of the manuscript. We would, however, like to note that most success rate statistics are artificially high because a preselection is made for the constructs that are most likely to express in a certain host that is evolutionarily close to the protein of origin. The broader point we make in this paragraph is that, given a particular host (eg yeast) and a random protein from any other source, our knowledge of the requirements for successful expression (and we here particularly focus on secretory expression) is limited and whether or not the protein will make it out successfully is still more a matter of trial and error rather than something we can currently consciously design."

At the same time the introduction contains a paragraph on impact of their method for assessment for recombinant protein production:

Line 53:

In practice, the absence of such an integrated picture of the secretory system is most apparent in the unpredictability of heterologous protein secretion. Indeed, four decades after the recombinant DNA revolution, obtaining detectable levels of functional protein in a particular heterologous host system, secretory or not, is still principally a process of trial and error. Such low and unpredictable expression success rates slow down progress in the many fields of basic and applied life science where recombinant protein production comes into play."

Taken together the comment in the response letter and the updated introduction do not go together very well. For recombinant protein production one wants to stay as close as possible to the true production host ie. Producing a human protein in a mammalian host. And I do not agree that removing statistics makes it more convincing. Instead, I would suggest to remove/tone down the recombinant protein production angle in the introduction and instead focus on how amino acid sequence influences the basic principles for yeast protein secretion.

Comment 3b

Response to evaluation of translatability of fragments in a cell-free system.
Response is no, which is fine. However, at the same time authors discuss line 437

“Our observation that more fragments with a high overall positive charge (Lys + Arg content) are not secreted could relate to translational ribosomal stalling at polybasic stretches”

This hypothesis could be tested in an IVT system and it does not agree with the response to comment 3b where it is stated that it would not be of relevance to the findings in the manuscript

Reviewer #3:

Remarks to the Author:

The authors have clarified several issues I raised and have made necessary corrections of scope. I am satisfied with their clarifications and toning down their language but my overall concern re the applicability and (low) significance of the work for recombinant protein production remains.

Manuscript NCOMMS-20-47011

"Massively parallel interrogation of protein fragment secretability using SECRiFY reveals features influencing secretory system transit" by Boone et al.

Response to reviewers

Dear reviewers,

We sincerely appreciate the time you have taken to consider a revised version of our manuscript. Reviewer #2 had some additional remarks, for which we have formulated our responses in *red italics* below.

Comment 2

The authors reply that

"We acknowledge that success rates can vary wildly across hosts and heterologous proteins, and this has indeed generally improved over the years since the Structural Genomics consortia. We omitted these statistics from the revised version of the manuscript. We would, however, like to note that most success rate statistics are artificially high because a preselection is made for the constructs that are most likely to express in a certain host that is evolutionarily close to the protein of origin. The broader point we make in this paragraph is that, given a particular host (eg yeast) and a random protein from any other source, our knowledge of the requirements for successful expression (and we here particularly focus on secretory expression) is limited and whether or not the protein will make it out successfully is still more a matter of trial and error rather than something we can currently consciously design."

At the same time the introduction contains a paragraph on impact of their method for assessment for recombinant protein production:

Line 53:

In practice, the absence of such an integrated picture of the secretory system is most apparent in the unpredictability of heterologous protein secretion. Indeed, four decades after the recombinant DNA revolution, obtaining detectable levels of functional protein in a particular heterologous host system, secretory or not, is still principally a process of trial and such low and unpredictable expression success rates slow down progress in the many fields of basic and applied life science where recombinant protein production comes into play."

Taken together the comment in the response letter and the updated introduction do not go together very well. For recombinant protein production one wants to stay as close as possible to the true production host ie. Producing a human protein in a mammalian host. And I do not agree that removing statistics makes it more convincing. Instead, I would suggest to remove/tone down the recombinant

protein production angle in the introduction and instead focus on how amino acid sequence influences the basic principles for yeast protein secretion.

Author response:

We partially acknowledge what this reviewer says, and we have substantially reformulated the introductory paragraph to which the reviewer refers, in order to increase consistency and to tone down the recombinant protein production angle and focus on what our experiments reveal (relationships between amino acid sequence of protein fragments and yeast protein secretion).

*However, we respectfully disagree with the reviewer's statement above that one wants to stay as close as possible to the native organism in recombinant protein production. Most biotechnological protein production in industry and academia alike is testament to the fact that this is simply not the case. *E. coli* and *Pichia pastoris* yeast are the two most frequently utilized expression hosts in the published literature and mainstays of both biopharmaceutical and industrial biotechnological protein production. For complex (multi-domain, disulfide-bonded, glycosylated,...) proteins such as antibodies and viral glycoproteins, mammalian or insect cells are at present often the best solution, and it is a goal of fundamental studies into the secretory system's workings to try and modify the secretory system of yeast so that it could proficiently produce such more complex proteins as well. This is met with increasing success, and we wish to maintain that studies such as those enabled by our SECRiFY methodology have the potential to gradually, step-by-step gain a deeper understanding in sequence/structure-secretability relationships for proteins in the yeast systems.*

New introductory paragraph accommodating the reviewer's remaining request:

*"The secretion of recombinant proteins by heterologous hosts has long been a popular alternative to cytoplasmic expression because of the more straightforward purification, even for proteins that are not naturally secreted or located in a membrane. However, obtaining detectable levels of functional recombinant protein secreted by a given heterologous host is still too often a process of trial and error. Predicting the compatibility between recombinant protein and secretory host, and the engineering of protein or host towards increased compatibility, require models of the relationship between amino acid or structural determinants and successful protein secretion. Arguably, the availability of large-scale protein secretion data will help to demystify which and why proteins fail to pass the secretory pathway. The screening of parallel constructs or variant libraries of a protein of interest to increase recombinant protein expression success rates has gained momentum¹⁴⁻¹⁷, but it is often focused on intracellular expression and more importantly, focusses on just a single target. More comprehensive strategies to assess heterologous expression across entire proteomes do exist, but have generally also been limited to intracellular expression in *E. coli*, small proteomes, and cumbersome clone-by-clone strategies¹⁸⁻²¹. Thus, new methods for measuring secretion in high throughput are needed."*

Comment 3b

Response to evaluation of translatability of fragments in a cell-free system. Response is no, which is fine. However, at the same time authors discuss line 437 “Our observation that more fragments with a high overall positive charge (Lys + Arg content) are not secreted could relate to translational ribosomal stalling at polybasic stretches”

This hypothesis could be tested in an IVT system and it does not agree with the response to comment 3b where it is stated that it would not be of relevance to the findings in the manuscript.

Author response: Yes, this hypothesis could, in principle, be tested in an IVT system, although the scale of experimentation to ascertain that this is the main mechanism behind this statistical pattern would be very large. This is just one of the features that transpire from the saliency maps of our neural networks, and the effect size of the contribution to the classification accuracy is very difficult to ascertain at this point. Hence, we anticipate that hundreds of designed sequences would need to be tested to reliably make a call on whether it is this mechanism or another one that causes the statistical pattern that more fragments with a high overall positive charge are not secreted. It is certainly an interesting lead for further studies and designing protein coding libraries through synthetic DNA is becoming a somewhat more affordable proposition, but we respectfully submit that the results of such work will form material for future publications that will stand by themselves. Moreover, it has come to our attention that the ribosomal stalling effect at polybasic stretches predominantly occurs on polyA stretches (leading to poly-lysines when translated), and that both polyA stretches in the DNA and Lysines in the polypeptide chain are needed to result in a peptidyl-transfer RNA conformation suboptimal for peptide bond formation (Chandrasekaran et al, Nat Struct Mol Biol, 2019). We have therefore removed that sentence from the updated version of the manuscript as there may well be other mechanisms at play (e.g. sticking to negatively charged phospholipids along the secretory pathway or to phosphomannan in the yeast cell wall, to name just two possibilities).